# Balancing Safety and Return: Region-based Reward Penalty over Action Chunks For Offline Safe RL

## Abstract

In-sample learning has emerged as a powerful paradigm that mitigates the Out-of-Distribution (OOD) issue, which leads to violations of safety constraints in offline safe reinforcement learning (OSRL). Existing approaches separately train reward and cost value functions, yielding *suboptimal* policies within the safe policy space. To address this, we propose the *Region-Based Reward Penalty over Action Chunks* (R2PAC), a novel method that trains $h$-step optimal value function within the safe policy space. By penalizing reward signals over action chunks that may potentially lead to unsafe transitions, our method: (1) integrates cost constraints into reward learning for constrained return maximization; (2) improves joint training stability by accelerating the convergence speed with unbiased multi-step value estimation; (3) effectively avoids unsafe states through temporally consistent behaviors. Extensive experiments on the DSRL benchmark demonstrate that our method outperforms state-of-the-art algorithms, achieving the highest returns in 13 out of 17 tasks while maintaining the normalized cost below a strict threshold in all tasks.

## 1 Introduction

Despite the potential to mitigate online safety risks, OSRL still inherits fundamental challenges from both safety guarantees and offline regularization. A major challenge in offline reinforcement learning (RL) is distributional shift (Levine et al., 2020; Fujimoto et al., 2019), wherein unseen state-action pairs are often erroneously overestimated to have unrealistic values (Fujimoto et al., 2019). This overestimation makes the policy preferentially select OOD actions during deployment. To address these challenges, existing offline RL works propose to constrain the learned policy close to the behavior policy or penalize the Q-values of OOD actions (Fujimoto et al., 2019; Kumar et al., 2020; An et al., 2021; Kumar et al., 2019; Fujimoto and Gu, 2021). However, such approaches may result in overly conservative policies (Mao et al., 2023). Another class of methods, in-sample learning (Kostrikov et al., 2021; Hansen-Estruch et al., 2023; Xu et al., 2023; Garg et al., 2023; Xiao et al., 2023) such as implicit Q-learning (IQL) (Kostrikov et al., 2021), offers a potential alternative that effectively avoids the OOD issue. In-sample learning approximates optimal values without querying the value function of any unseen actions. In addition, its decoupled value function and policy learning processes provide additional feasibility to various parameterized actors (Hansen-Estruch et al., 2023; Zheng et al., 2024; Wang et al., 2023).

A major challenge imposed by safety constraints is the necessity of preventing unsafe actions that could result in catastrophic outcomes (García et al., 2015; Ray et al., 2019; Brunke et al., 2022; Andersen et al., 2020; Shi et al., 2021). A magnitude of safe RL approaches, such as primal-dual methods (Wu et al., 2024; Chow et al., 2018) and reward penalty techniques (Thomas et al., 2021; Araújo and Braga, 1998), fail to consistently satisfy constraints despite their foundation in constrained optimization, resulting in significant training performance instability. Feasibility analysis (Fisac et al., 2019; Yu et al., 2022) can effectively enforce constraints, though its strong focus on constraint satisfaction may result in overly conservative policies.

While the offline in-sample method IQL shows potential for incorporating safety constraints (Koirala et al., 2025; Zheng et al., 2024), it faces significant challenges in guaranteeing safety. In the exist-

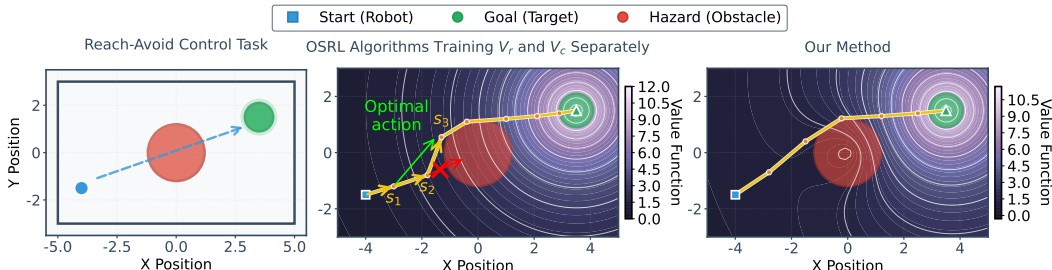

Figure 1: An illustrative example of a reach-avoid control task. Left: The robot aims to reach the goal while avoiding hazards. Middle: Under decoupled value functions, the robot at $s_1$ myopically chooses the action maximizing $V_r$ (yellow arrow to $s_2$). At $s_2$, the $V_r$-maximizing action is unsafe (red arrow), forcing a safe alternative (yellow arrow to $s_3$) by $V_c$. However, the globally optimal trajectory requires selecting the green arrow at $s_1$. Right: In contrast, incorporating a reward penalty into IQL encourages a coherent value function that balances safety and task achievement, resulting in a shorter and more efficient hazard avoidance path toward the goal.

ing research, the independent training of reward and cost value functions can lead to the extracted policy diverging from the optimal policies implied by each value function, thereby risking safety failures or performance degradation. As shown in Figure 1, the separate training of reward and cost value functions leads them to provide conflicting policy recommendations, forcing the policy onto a suboptimal trajectory. In contrast, jointly training the value functions enables the agent to foresee potential risks while pursuing reward maximization.

To address this issue, different from existing in-sample OSRL methods with separate training of the reward and cost value function, we propose the *Region-Based Reward Penalty over Action Chunks* (R2PAC), a novel in-sample OSRL method that is the first approach to enable IQL to directly learn a single value function whose corresponding implicit policy is optimal within the safe policy space. Our method introduces two effective extensions to the IQL algorithm, achieving strong performance in both reward maximizing and constraint satisfaction.

The first innovation is on a new region-based reward-penalized update within the IQL framework, which extends the reward value function of IQL to incorporate safety constraints. The key breakthrough is that a straightforward yet powerful region-specific reward penalty enables the standard IQL framework to learn a value function that corresponds to the constrained optimal policy. Specifically, a key step is to learn an optimal cost value function, through which we accurately identify safe regions and establish a well-defined region that guides our targeted penalty design. Subsequently, we penalize the reward signal for transitions into unsafe areas. Theoretical analysis shows that this penalization ensures policy safety without sacrificing optimality, which is supported by our argument that an unsafe policy is always inferior to a safe one under our penalization. As a result, safe policy extraction can be simplified using only a single value function.

We propose, for the first time, integrating action chunking, a technique widely employed in imitation learning (Black et al., 2025; Li et al., 2025), into in-sample learning to stabilize the training of value functions and to enhance its risk anticipation capability. First, we identify for the first time that, by leveraging multi-step backpropagation to mitigate the impact of temporal difference (TD) errors, action chunking helps to address the overfitting problem, a common issue in in-sample learning methods (Chen et al., 2025; Xu et al., 2023; Garg et al., 2023). Furthermore, since our reward value function is trained in dependency on the cost value function, action chunking helps stabilize the training of the reward value function compared to single-step updates. Secondly, by predicting action sequences over the next $h$ steps, our method significantly improves the agent's danger avoidance capability. In contrast to single-step prediction, our approach equips the actor with implicit long-horizon foresight, enabling earlier anticipation of potential hazards.

The proposed method is straightforward to implement and highly transferable to most OSRL algorithms, as it requires no additional modules and uses a simple multi-layer perceptron (MLP) as the network backbone. Indeed, merely by changing to rely on a single value function and incorporat-

ing action chunking, we can avoid the challenge faced by existing in-sample learning methods in balancing the contributions of reward and cost value functions.

Extensive experimental results demonstrate that our method outperforms other OSRL algorithms on the DSRL benchmark (Liu et al., 2023a). Adopting only a single value function for policy extraction, our method satisfies safety constraints across all 17 tasks and achieves the highest reward in 13 of them.

## 2 RELATED WORK

**Offline Safe RL.** Safe offline RL learns policies from fixed datasets under safety constraints, commonly formulated as a Constrained Markov Decision Process (CMDP) (Altman, 2021). Existing methods often combine Lagrangian-based safe RL techniques with offline algorithms (Kostrikov et al., 2021; Xu et al., 2022; Zheng et al., 2024). Representative works include CPQ (Xu et al., 2022), which penalizes unsafe and OOD actions but may harm generalization, and FISOR (Zheng et al., 2024), which enforces strict safety via Hamilton–Jacobi reachability. Alternative sequence modeling approaches (Liu et al., 2023b; Lin et al., 2023; Zhang et al., 2023; Ze Gong, 2025) exist but are sensitive to data quality. Approaches such as LSPC (Koirala et al., 2025) rely on custom parameterized actors for safety, which limits their scalability to other designs and leads to poor performance on difficult tasks. Recent works address offline safe RL from data distribution perspectives. OASIS (Yihang Yao, 2024) shapes dataset distributions toward safe and high-reward regions to mitigate dataset mismatch, while Ze Gong (2025) directly learns trajectory-generating policies. However, these methods cannot adapt to varying cost constraints. Newer approaches like CAPS (Yassine Chemingui, 2025) switch between policies at test-time, and CCAC (Zijian Guo, 2025) models state-action distributions for zero-shot constraint adaptation. Yet, they fail to handle divergence between policy and value, resulting in suboptimal performance.

**Action Chunking.** Action chunking has been widely used in imitation learning to improve policy robustness and handle non-Markovian behavior in offline datasets (Li et al., 2025; Black et al., 2025; Liu et al., 2025; Bharadhwaj et al., 2024; George and Farimani, 2023). This approach predicts and executes action sequences in an open-loop manner, requiring a powerful parameterized actor. Learning such chunked policies often relies on expressive generative models, including diffusion (Sohl-Dickstein et al., 2015; Ho et al., 2020; Song et al., 2020; Croitoru et al., 2023) and flow matching (Frans et al., 2025; Gao et al., 2025; Lipman et al., 2024; Chen and Lipman, 2023; Lipman et al., 2022). In this work, we repurpose action chunking to regularize the value function and promote safety through temporally consistent actions in in-sample learning.

## 3 PRELIMINARY

**Offline Safe RL.** Safe RL is typically formulated as a CMDP (Altman, 2021), which is defined by a tuple $\mathcal{M} := (\mathcal{S}, \mathcal{A}, P, r, c, \gamma)$. This tuple comprises a state space $\mathcal{S}$, an action space $\mathcal{A}$, a transition dynamics $P : \mathcal{S} \times \mathcal{A} \to \Delta(\mathcal{S})$, a reward function $r : \mathcal{S} \times \mathcal{A} \to \mathbb{R}$, a cost function $c : \mathcal{S} \times \mathcal{A} \to \mathbb{R}$, and a discount factor $\gamma \in [0, 1]$. The objective of safe RL is to find a policy $\pi(a|s)$ to maximize the expected cumulative rewards while satisfying the safety constraint, i.e. $\max_\pi \mathbb{E}_{s_0 \sim \rho_0, a_t \sim \pi, s_{t+1} \sim P} \left[ \sum_{t=0}^{T-1} \gamma^t r(s_t, a_t) \right]$, $s.t. \mathbb{E}_{s_0 \sim \rho_0, a_t \sim \pi, s_{t+1} \sim P} \left[ \sum_{t=0}^{T-1} \gamma^t c(s_t, a_t) \right] \leq \ell$, where $\rho_0$ denotes the distribution of initial states, $T$ is the trajectory length, and $\ell \geq 0$ is the predefined cost limit.

In offline RL, the objective is to optimize the safe RL objective with a previously collected dataset $\mathcal{D} = \{(s_t^{(i)}, a_t^{(i)}, r_t^{(i)}, c_t^{(i)}, s_{t+1}^{(i)})_{t=0}^{T-1}\}_{i=0}^{N-1}$, which contains a total of $N$ trajectories. Existing offline safe RL methods typically solve the problem in the following form:

$$\max_\pi \quad \mathbb{E}_{s_t, a_t} \left[ \sum_{t=0}^{T-1} \gamma^t r(s_t, a_t) \right] \quad s.t. \quad \mathbb{E}_{s_t, a_t} \left[ \sum_{t=0}^{T-1} \gamma^t c(s_t, a_t) \right] \leq \ell; \quad D(\pi||\pi_\beta) \leq \epsilon, \quad (1)$$

where $\pi_\beta$ is the underlying behavioral policy of the offline dataset, $D(\pi||\pi_\beta)$ is a divergence term to prevent the learned policy $\pi$ shift form $\pi_\beta$.

**Implicit Q Learning.** IQL (Kostrikov et al., 2021) addresses the distribution shift issue by approximating value functions only on in-distribution data. This method is achieved through an asymmetric

$\ell_2$ loss (i.e., expectile regression). The losses for parameterized Q-function and state value function in IQL are as follows:

$$\mathcal{L}_Q(\theta) = \mathbb{E}_{(s,a,s')\sim D}\left[(r(s,a) + \gamma V_\psi(s') - Q_\theta(s,a))^2\right],\tag{2}$$

$$\mathcal{L}_V(\psi) = \mathbb{E}_{(s,a)\sim D}\left[L_2^\tau(Q_{\hat{\theta}}(s,a) - V_\psi(s))\right],\tag{3}$$

where $L_2^\tau(u) = |\tau - \mathbb{I}(u < 0)|u^2$. For policy extraction, IQL uses Advantage Weighted Regression (AWR) (Peng et al., 2019):

$$\mathcal{L}_\pi(\phi) = \mathbb{E}_{(s,a)\sim D}\left[\exp(\alpha(Q_{\hat{\theta}}(s,a) - V_\psi(s)))\log\pi_\phi(a|s)\right],\tag{4}$$

where $\alpha \in [0, \infty]$ is the temperature parameter.

## 4 METHODS

In this section, we introduce the formulation of the proposed *region-based reward penalty over action chunks*, which yields a single value function to achieve both reward maximization and constraint satisfaction. Our approach comprises three key components: (1) the derivation of a region-based reward penalty mechanism, (2) the adaptation of action chunking to stabilize value function training, and (3) a simple yet effective policy extraction method based on flow-matching.

### 4.1 PENALIZED REWARD VALUE FUNCTION

To address safety requirements in offline RL, we first introduce a penalized value function. Our objective is to learn a policy that consistently satisfies safety constraints, relying solely on in-sample data.

We first separate the entire state space $\mathcal{S}$ into the *safe region* ($\mathcal{S}_{\text{safe}}^\pi$) and the *unsafe region* ($\mathcal{S}_{\text{unsafe}}^\pi$). Following the conventions established in prior work (Fisac et al., 2019; Zheng et al., 2024; Thomas et al., 2021), we have the following definition:

**Definition 1** *The safe region and the unsafe region under a policy $\pi$ are defined as:*

$$\mathcal{S}_{safe}^\pi := \{s \in \mathcal{S}|V_c^\pi \le \ell\} \quad and \quad \mathcal{S}_{unsafe}^\pi := \mathcal{S}/\mathcal{S}_{safe}^\pi.\tag{5}$$

We denote $\mathcal{S}_{\text{safe}}^*$ as the largest safe region, which is induced by the policy $\pi_c^* := \arg\min_{\pi\in\Pi} V_c^\pi(s)$. By definition, for any state within $\mathcal{S}_{\text{safe}}^*$, there exists at least one policy that satisfies the safety constraints from that state onward. Conversely, if a state belongs to $s \in \mathcal{S}_{\text{unsafe}}^* := \mathcal{S}/\mathcal{S}_{\text{safe}}^*$, any trajectory starting from it will eventually violate the safety threshold $\ell$, since even the safest feasible policy $\pi_c^*$ fails to satisfy the constraint at $s$. Accordingly, we define the safe policy space as follows:

**Definition 2** *The safe policy space is defined as:*

$$\Pi_{safe} := \{\pi \in \Pi|V_c^\pi(\rho_0) \le \ell\},\tag{6}$$

*where $\rho_0$ is the initial state distribution.*

**Region-Based Reward Penalty Framework.** Inspired by Thomas et al. (2021), we adopt the reward penalty framework to ensure safety. For a transition $(s, a, s', r)$, we transform it to:

$$(s, a, s', r) = \begin{cases} (s, a, s', r) & \text{If } s' \in \mathcal{S}_{\text{safe}}^* \\ (s, a, s_a, -C) & \text{If } s' \in \mathcal{S}_{\text{unsafe}}^* \end{cases},\tag{7}$$

where $C \in \mathbb{R}$ is a penalty constant for unsafe transitions, and $s_a$ denotes an absorbing state: any action taken in $s_a$ returns to $s_a$ with reward $-C$. Under this formulation, any unsafe action that would lead to an unsafe next state is effectively redirected to $s_a$. Intuitively, provided that $C$ is sufficiently large, any policy that maximizes cumulative reward from a safe state $s$ is guaranteed to avoid entering the unsafe region. We denote the reward value function learned with this transformation as $\overline{V}_r$. To formalize this intuition, we present the following proposition:

**Proposition 1** *Assume that the safe policy space $\Pi_{safe}$ is non-empty for any $\ell \geq 0$. Then for any safe policy $\pi_1 \in \Pi_{safe}$ and any unsafe space $\pi_2 \in \Pi/\Pi_{safe}$, the value functions satisfy*

$$\overline{V}_r^{\pi_1}(\rho_0) > \overline{V}_r^{\pi_2}(\rho_0) \tag{8}$$

*under the condition*

$$C > \frac{\sum_{t=0}^{T-1} \gamma^t (r_{\max} - r_{\min})}{\ell}, \tag{9}$$

*where $r_{\max}$ and $r_{\min}$ denote maximum and minimum of the reward.*

The above discussion ensures that, starting from a safe state, a policy maximizing $\overline{V}_r$ will, in theory, never enter the unsafe region. Moreover, we demonstrate that our penalty framework also preserves optimality when extended to hard constraints. The proof and corresponding discussion are provided in Appendix A.1 and Appendix A.2.

**Learning the Safe and Reward Value Functions.** We employ IQL to learn the optimal cost value functions $Q_c^*, V_c^*$ using

$$\mathcal{L}_{Q_c} = \mathbb{E}_{(s,a,s') \sim D} \Big[ (c(s,a) + \gamma V_c(s') - Q_c(s,a))^2 \Big], \tag{10}$$

$$\mathcal{L}_{V_c} = \mathbb{E}_{(s,a) \sim D} \Big[ L_2^\tau (V_c(s,a) - Q_c(s)) \Big], \tag{11}$$

Based on $V_c^*$, we derive the largest safe region $\mathcal{S}_{\text{unsafe}}^*$. Additionally, we apply the same IQL framework to learn the optimal reward value function using the converted transition data:

$$\mathcal{L}_{\overline{Q}_r} = \mathbb{E}_{(s,a,s') \sim D} \Big[ (\mathbb{I}(V_c(s') \leq \ell)(r(s,a) + \gamma \overline{V}_r(s')) + \mathbb{I}(V_c(s') > \ell)\overline{C} - \overline{Q}_r(s,a))^2 \Big], \tag{12}$$

$$\mathcal{L}_{\overline{V}_r} = \mathbb{E}_{(s,a) \sim D} \Big[ L_2^\tau (\overline{Q}_r(s,a) - \overline{V}_r(s)) \Big], \tag{13}$$

where $\overline{C} = \frac{-C}{1-\gamma} = \sum_{t=0}^{\infty} \gamma^t(-C)$ denotes the value of the absorbing state $s_a$, and $\mathbb{I}(\cdot)$ represents the indicator function. To extract a policy that maximizes cumulative reward while satisfying safety constraints, it suffices to maximize the penalized reward value function $\overline{V}_r$.

## 4.2 Penalized Value Function with Action Chunking

Although Section 4.1 offers a theoretically justified framework for safety guarantees, the training of the penalized reward value function $\overline{V}_r$ can exhibit instability caused by the dependency of $\overline{V}_r$ on $V_c$, where bootstrap errors propagate across both temporal horizon and value functions. To address this, we integrate an action chunking technique into the IQL framework for value function learning.

**Chunked Q-function.** We generalize the critic training to operate over a horizon of $h$ consecutive actions:

$$Q_r(s_t, a_{t:t+h}) = \sum_{k=t}^{t+h-1} \gamma^k r_k + \gamma^h \mathbb{E}_{s_{t+h}}[V_r(s_{t+h})],$$

$$Q_c(s_t, a_{t:t+h}) = \sum_{k=t}^{t+h-1} \gamma^k c_k + \gamma^h \mathbb{E}_{s_{t+h}}[V_c(s_{t+h})], \tag{14}$$

where $a_{t:t+h} = \{a_t, a_{t+1}, \cdots, a_{t+h-1}\}$.

The chunked Q-function described above exhibits a structural similarity to the uncorrected $n$-step return (Kozuno et al., 2021; Hessel et al., 2018), which introduces bias by conditioning only on the first action while using rewards from a sequence of actions. In contrast, our method remains unbiased, as the Q-function explicitly takes the entire action sequence as input.

**Alleviate Overfitting in IQL.** In IQL, the use of a large $\tau$ (close to 1) in the asymmetric loss function leads to better approximate the maximum value function. However, this approach can cause the value function to overfit to overestimated Q-values originating from bootstrap errors. We present the first analysis of Q-value overestimation in IQL from a temporal horizon perspective, and introduce an action chunking technique to mitigate the effect of bootstrap errors. The overestimation is primarily due to the propagation of bootstrap errors during the standard 1-step temporal difference backups. This propagation results in cumulative error amplification over extended temporal horizons.

Unlike previous studies that mitigate overfitting via value ensembles in 1-step IQL (Chen et al., 2025), our action chunking technique reduces approximation error and enhances numerical stability by decreasing error propagation frequency without a huge number of value networks. The use of $h$-step returns propagates value $h$ times faster, accelerating convergence by reducing error propagation steps, thereby significantly improving numerical stability compared to 1-step IQL.

To demonstrate the effectiveness of action chunking in stabilizing training, we train vanilla IQL on the *BallRun* task from the DSRL benchmark (Liu et al., 2023a). As illustrated in Figure 2, vanilla IQL with a chunk length

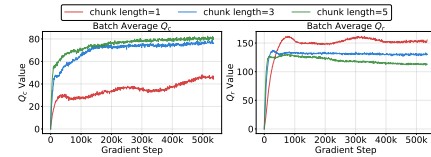

Figure 2: Batch-averaged $Q_r$ and $Q_c$ values obtained during training with vanilla IQL on the Bullet-Safety-Gym *BallRun* task across various chunk lengths.

of 1 exhibits slow convergence and unstable Q-values. In contrast, increasing the chunk length leads to faster convergence of the batch-averaged Q-values and smoother training curves, indicating significantly improved stability throughout training.

**Action Chunked Penalized Value Function.** Our action chunking technique serves as a drop-in modification and can be straightforwardly integrated with IQL. We employ a loss formulation similar to Eq. 12 and 13 to train the $h$-step penalized reward value function $\overline{Q}_r$ and state value function $\overline{V}_r$:

$$\mathcal{L}_{\overline{Q}_r} = \mathbb{E}_{(s_t, a_{t:t+h}, s_{t+h}) \sim D}\Big[\Big(\mathbb{I}(V_c(s_{t+h}) \le \ell)\Big(\sum_{k=0}^{h-1} \gamma^k r_{t+k} + \gamma^h \overline{V}_r(s_{t+h})\Big)+$$

$$\mathbb{I}(V_c(s_{t+h}) > \ell)\overline{C} - \overline{Q}_r(s_t, a_{t:t+h})\Big)^2\Big], \tag{15}$$

$$\mathcal{L}_{\overline{V}_r} = \mathbb{E}_{(s_t, a_{t:t+h}) \sim D}\Big[L_2^\tau\Big(\overline{Q}_r(s_t, a_{t:t+h}) - \overline{V}_r(s_t)\Big)\Big]. \tag{16}$$

This loss extends the penalized Q-loss to $h$ steps. To ensure consistency, we also train an $h$-step policy $\pi(a_{t:t+h}|s_t)$ designed to generate a sequence of $h$ consecutive actions. Since maximizing $\overline{Q}_r$ is equivalent to solving a constrained optimization problem, the policy is trained by directly maximizing $\overline{Q}_r$. Analogous to 1-step IQL, we state the following proposition:

**Proposition 2** *In the limit as $\tau \to 1$, $h$-step IQL converges to the optimal value function of an induced MDP where each transition represents $h$ steps in the original MDP.*

Proposition 2 establishes that combining IQL with action chunking is equivalent to learning the optimal value function in an induced MDP. This finding enables the seamless integration of action chunking into our reward-penalty framework, ensuring interpretability. The proof is provided in Appendix A.3.

### 4.3 Practical Implement

Although the optimal policy maximizes $\overline{V}_r$ to avoid entering unsafe regions, value function approximation errors may still cause the agent to deviate into unsafe states during execution. Once in such states, relying solely on $\overline{V}_r$ provides no mechanism for identifying escape paths back to safety. To overcome this limitation, which is further discussed in Section 5.2, we introduce decoupled objectives for safe and unsafe states. Our approach leverages both $\overline{V}_r$ and the cost value function $V_c$ to guide policy recovery and ensure safety:

$$s \in \mathcal{S}_{\text{safe}} : \max_\pi \mathbb{E}_s\Big[\overline{V}_r^\pi(s)\Big], \qquad s \in \mathcal{S}_{\text{unsafe}} : \max_\pi \mathbb{E}_s\Big[-V_c^\pi(s)\Big]. \tag{17}$$

In the spirit of preserving simplicity and efficiency, we aim for a simple method for policy extraction. We adopt a novel framework based on flow-matching models (Gao et al., 2025; Lipman et al., 2024) utilizing classifier-free guidance (CFG) to generate actions from complex multi-modal distributions (Frans et al., 2025).

We instantiate a single flow-matching network to serve as both the conditional and unconditional policy. The policy is modeled by a velocity field $v_\theta$, conditioned on a partially-noised action $a_{t:t+h}^i$,

noise scale $i$, current state $s_t$, and an optimality variable $o \in \{\emptyset, 0, 1\}$. The optimality variable is instantiated as follows:

$$s_t \in \mathcal{S}_{\text{safe}} : \qquad\qquad\qquad\qquad\qquad s_t \in \mathcal{S}_{\text{unsafe}} :$$

$$o = \begin{cases} 1 & \text{if} \quad \overline{A}_r = \overline{Q}_r - \overline{V}_r \geq 0 \\ 0 & \text{if} \quad \overline{A}_r = \overline{Q}_r - \overline{V}_r \leq 0 \end{cases}, \qquad o = \begin{cases} 1 & \text{if} \quad A_c = Q_c - V_c \leq 0 \\ 0 & \text{if} \quad A_c = Q_c - V_c > 0 \end{cases}. \tag{18}$$

$v_\theta$ is trained via the following loss function:

$$\mathcal{L}_v = \mathbb{E}_{(s_t, a_{t:t+h}) \sim D}\Big[||v_\theta(a_{t:t+h}^i, i, s_t, o) - (a_{t:t+h} - a_{t:t+h}^0)||^2\Big], \tag{19}$$

$$a_{t:t+h}^i = (1-i)a_{t:t+h}^0 + i a_{t:t+h}, \tag{20}$$

and where the noise scale $i$ is sampled uniformly from $[0, 1]$, and $a_{t:t+h}^0 \sim \mathcal{N}(0, 1)$ is Gaussian noise. When executing, the noised actions are denoised by:

$$\hat{v}^i = v_\theta(a_{t:t+h}^i, i, s_t, \emptyset) + \omega(v_\theta(a_{t:t+h}^i, i, s_t, o) - v_\theta(a_{t:t+h}^i, i, s_t, \emptyset)), \tag{21}$$

where $\omega$ is the guidance scale. At each timestep, we generate a sequence of $h$ actions but execute only the first one. Given that most tasks in the DSRL benchmark involve infinite horizons, defining an appropriate safety threshold is challenging. To address this, we adopt the $\alpha$-quantile of the $V_c$ values from the dataset as the safety threshold. To further enhance safety, we employ rejection sampling (Hatch et al., 2024; Hansen-Estruch et al., 2023; Chen et al., 2023) to select the action with lowest $Q_c$. Please see Appendix C, Appendix D, and Appendix E.5 for more details.

# 5 EXPERIMENTS

## 5.1 EVALUATION ON DSRL BENCHMARK

**Datasets and Metrics.** We evaluate the proposed method on Safety-Gymnasium (Ray et al., 2019) and Bullet-Safety (Gronauer, 2022) tasks within the DSRL benchmark (Liu et al., 2023a), comparing against state-of-the-art safe offline RL algorithms. Performance is measured using normalized return and normalized cost. The normalized return is computed as $R = (R_\pi - R_{\min})/(R_{\max} - R_{\min})$, where $R_\pi$ is the return of the current policy, and $R_{\max}$ and $R_{\min}$ are the maximum and minimum returns in the dataset. The normalized cost is given by $C = C_\pi/\kappa$, where $\kappa > 0$ is the cost threshold. Following FISOR (Zheng et al., 2024), we treat safety as the primary evaluation criterion and aim to maximize reward only when the safety constraint is satisfied. We set $\kappa = 10$ for Safety-Gymnasium and $\kappa = 5$ for Bullet-Safety tasks, which are used in testing the algorithms aiming to achieve the hard constraints and are hard to achieve in other OSRL algorithms.

**Baselines.** The baseline algorithms include: i) BC: Behavior cloning; ii) CPQ (Xu et al., 2022): a constrained Q-learning approach that penalizes OOD actions as unsafe; iii) COptiDICE (Lee et al., 2022): a Lagrangian method based on distribution correction estimation (DICE), extending OptiDICE (Lee et al., 2021) for offline safe RL; iv) CDT (Liu et al., 2023b): a Constrained Decision Transformer that infers future costs; v) TREBI (Lin et al., 2023): a cost-budget inference method leveraging the Diffuser (Janner et al., 2022; Ajay et al., 2022) for real-time safe decision-making; vi) FISOR (Zheng et al., 2024): a feasibility guided method combined with diffusion policies; vii) CAPS(IQL) (Yassine Chemingui, 2025): a wrapper framework that dynamically switches between reward-maximizing and cost-minimizing policies.

**Main Results.** The evaluation results are presented in Table 1. We primarily adopt the evaluation metrics reported in FISOR (Zheng et al., 2024), except for the evaluation of the *Point* agent on Safety-Gymnasium and baseline algorithms CAPS, which is conducted independently as it is not in the source. Our proposed method demonstrates significant performance improvements over all existing baselines. The first five baseline algorithms exhibit substantial constraint violations under strict safety requirements, as a significant gap exists between their theoretical guarantees and practical implementation. Although FISOR explicitly incorporates hard constraints and achieves satisfactory constraint satisfaction, it tends to be overly conservative in many tasks, resulting in comparatively low returns. In contrast, our method maximizes the optimal value function within the safe policy space. Despite employing a more lenient cost metric $V_c$ compared to the feasibility-oriented objective, our approach still satisfies all strict cost constraints. The experimental results clearly demonstrate that our method outperforms all baselines in terms of reward attainment, achieving the highest reward in most of the tasks, while consistently keeping the normalized cost below 1 across all tasks.

Table 1: Normalized DSRL (Liu et al., 2023a) benchmark results. ↑ means the higher the better. ↓ means the lower the better. Each value is averaged over 20 evaluation episodes and 3 random seeds. Gray: Unsafe agents. **Bold**: Safe agents whose normalized cost is smaller than 1. Red: Safe agents with the highest reward. Blue: Safe agents with the second highest reward.

| Task | BC reward ↑ | BC cost ↓ | CDT reward ↑ | CDT cost ↓ | CPQ reward ↑ | CPQ cost ↓ | COptiDICE reward ↑ | COptiDICE cost ↓ | TREBI reward ↑ | TREBI cost ↓ | FISOR reward ↑ | FISOR cost ↓ | CAPS(IQL) reward ↑ | CAPS(IQL) cost ↓ | R2PAC(ours) reward ↑ | R2PAC(ours) cost ↓ |
|---|---|---|---|---|---|---|---|---|---|---|---|---|---|---|---|---|
| PointButton1 | 0.10 | 3.36 | 0.41 | 9.72 | 0.26 | 8.06 | 0.08 | 4.59 | 0.10 | 2.92 | **0.14** | **0.33** | -0.01 | 1.23 | **0.15** | **0.74** |
| PointButton2 | 0.29 | 8.20 | 0.23 | 5.56 | 0.34 | 11.29 | 0.18 | 5.74 | 0.25 | 3.30 | **0.13** | **0.65** | 0.13 | 2.91 | **0.16** | **0.63** |
| PointPush1 | 0.20 | 3.09 | 0.21 | 2.02 | 0.01 | 1.51 | 0.13 | 2.91 | 0.03 | 1.02 | **0.23** | **0.32** | 0.11 | 1.01 | **0.19** | **0.36** |
| PointPush2 | 0.18 | 4.30 | 0.17 | 2.23 | 0.08 | 17.28 | 0.04 | 4.59 | 0.14 | 1.19 | **0.12** | **0.64** | 0.09 | 1.87 | **0.20** | **0.25** |
| PointGoal1 | 0.64 | 4.44 | 0.52 | 2.69 | **0.52** | **0.50** | 0.53 | 4.75 | 0.32 | 1.72 | **0.16** | **0.30** | 0.21 | 0.96 | **0.36** | **0.20** |
| PointGoal2 | 0.52 | 8.16 | 0.30 | 2.64 | 0.03 | 6.38 | 0.42 | 6.55 | 0.34 | 5.87 | **0.10** | **0.07** | 0.23 | 1.33 | **0.48** | **0.95** |
| CarButton1 | 0.01 | 6.19 | 0.17 | 7.05 | 0.22 | 40.06 | -0.16 | 4.63 | 0.07 | 3.75 | **-0.02** | **0.26** | -0.07 | 1.27 | **0.15** | **0.96** |
| CarButton2 | -0.10 | 4.47 | 0.23 | 12.87 | 0.08 | 19.03 | -0.17 | 3.40 | **-0.03** | **0.97** | **0.01** | **0.58** | -0.08 | 1.36 | **0.07** | **0.25** |
| CarPush1 | 0.21 | 1.97 | 0.27 | 2.12 | **0.08** | **0.77** | 0.21 | 1.28 | 0.26 | 1.03 | **0.28** | **0.28** | 0.17 | 0.45 | **0.29** | **0.36** |
| CarPush2 | 0.11 | 3.89 | 0.16 | 4.60 | -0.03 | 10.00 | 0.10 | 4.55 | 0.12 | 2.65 | **0.14** | **0.89** | 0.04 | 1.61 | **0.15** | **0.52** |
| CarGoal1 | 0.35 | 1.54 | 0.60 | 3.15 | 0.33 | 4.93 | 0.43 | 2.81 | 0.41 | 1.16 | **0.49** | **0.83** | 0.25 | 0.95 | **0.42** | **0.57** |
| CarGoal2 | 0.22 | 3.30 | 0.45 | 6.05 | 0.10 | 6.31 | 0.19 | 2.83 | 0.13 | 1.16 | **0.06** | **0.33** | 0.14 | 2.73 | **0.32** | **0.67** |
| AntVel | 0.99 | 12.19 | **0.98** | **0.91** | -1.01 | 0.00 | 1.00 | 10.29 | 0.31 | 0.00 | **0.89** | **0.00** | 0.85 | 0.13 | **0.92** | **0.88** |
| HalfCheetahVel | 0.97 | 17.93 | **0.97** | **0.55** | 0.08 | 2.56 | 0.43 | 0.00 | 0.87 | 0.23 | **0.89** | **0.00** | 0.90 | 0.19 | **0.95** | **0.13** |
| SwimmerVel | 0.38 | 2.98 | 0.67 | 1.47 | 0.31 | 11.58 | 0.58 | 23.64 | 0.42 | 1.31 | -0.04 | 0.00 | **0.53** | 0.34 | **0.55** | **0.12** |
| **SafetyGym Average** | 0.34 | 5.73 | 0.42 | 4.24 | 0.09 | 9.35 | 0.27 | 5.50 | 0.25 | 1.88 | **0.24** | **0.37** | 0.23 | 1.22 | **0.35** | **0.50** |
| AntRun | 0.73 | 11.73 | 0.70 | 1.88 | **0.00** | **0.00** | 0.62 | 3.64 | 0.63 | 5.43 | **0.45** | **0.03** | 0.49 | 2.71 | **0.62** | **0.53** |
| BallRun | 0.67 | 11.38 | **0.32** | **0.45** | 0.85 | 13.67 | 0.55 | 11.32 | 0.29 | 4.24 | **0.18** | **0.00** | 0.10 | 0.16 | **0.30** | **0.00** |
| CarRun | 0.96 | 1.88 | 0.99 | 1.10 | 1.06 | 10.49 | 0.92 | 0.00 | 0.97 | 1.01 | **0.73** | **0.14** | **0.96** | 0.37 | **0.93** | **0.57** |
| DroneRun | 0.55 | 5.21 | **0.58** | **0.30** | 0.02 | 7.95 | 0.72 | 13.77 | 0.59 | 1.41 | **0.30** | **0.55** | 0.19 | 2.01 | **0.59** | **0.26** |
| AntCircle | 0.65 | 19.45 | 0.48 | 7.44 | **0.00** | **0.00** | 0.18 | 13.41 | 0.37 | 2.50 | **0.20** | **0.00** | **0.30** | 0.00 | **0.32** | **0.23** |
| BallCircle | 0.72 | 10.02 | 0.68 | 2.10 | 0.40 | 4.37 | 0.70 | 9.06 | 0.63 | 1.89 | **0.34** | **0.00** | **0.38** | 0.00 | **0.51** | **0.49** |
| CarCircle | 0.65 | 11.16 | 0.71 | 2.19 | 0.49 | 4.48 | 0.44 | 7.73 | **0.49** | **0.73** | 0.40 | 0.11 | 0.42 | 0.01 | **0.52** | **0.11** |
| DroneCircle | 0.82 | 13.78 | 0.55 | 1.29 | -0.27 | 1.29 | 0.24 | 2.19 | 0.54 | 2.36 | **0.48** | **0.00** | 0.32 | 0.00 | **0.52** | **0.32** |
| **BulletGym Average** | 0.72 | 10.58 | 0.63 | 2.09 | 0.32 | 5.28 | 0.55 | 7.64 | 0.56 | 2.45 | 0.39 | 0.10 | **0.40** | 0.66 | **0.54** | **0.31** |
| easysparse | 0.32 | 4.73 | **0.05** | **0.10** | -0.06 | 0.24 | 0.94 | 18.21 | 0.26 | 6.22 | **0.38** | **0.53** | 0.77 | 9.74 | **0.42** | **0.07** |
| easymean | 0.22 | 2.68 | **0.27** | **0.24** | -0.06 | 0.24 | 0.74 | 14.81 | 0.19 | 4.85 | **0.38** | **0.25** | 0.00 | 0.21 | **0.38** | **0.02** |
| easydense | 0.20 | 1.70 | 0.43 | 2.31 | -0.06 | 0.29 | 0.60 | 11.27 | 0.26 | 5.81 | **0.36** | **0.25** | 0.31 | 1.93 | **0.38** | **0.05** |
| mediumsparse | 0.53 | 1.74 | 0.26 | 2.20 | -0.08 | 0.18 | 0.64 | 7.26 | 0.06 | 1.70 | **0.42** | **0.22** | 0.36 | 0.57 | **0.86** | **0.21** |
| mediummean | 0.66 | 2.94 | 0.28 | 2.13 | -0.08 | 0.28 | 0.73 | 8.35 | 0.20 | 1.90 | **0.39** | **0.08** | 0.38 | 0.48 | **0.84** | **0.28** |
| mediumdense | 0.65 | 3.79 | **0.29** | **0.77** | -0.08 | 0.20 | 0.91 | 9.52 | 0.03 | 1.18 | **0.49** | **0.44** | 0.22 | 0.16 | **0.82** | **0.15** |
| hardsparse | 0.28 | 1.98 | **0.17** | **0.47** | -0.04 | 0.28 | 0.34 | 7.34 | 0.00 | 0.82 | **0.30** | **0.01** | 0.17 | 0.76 | **0.40** | **0.03** |
| hardmean | 0.34 | 3.76 | 0.28 | 3.32 | -0.05 | 0.24 | 0.36 | 7.51 | 0.16 | 4.91 | **0.26** | **0.09** | 0.30 | 3.62 | **0.38** | **0.03** |
| harddense | 0.40 | 5.57 | 0.24 | 1.49 | -0.04 | 0.24 | 0.42 | 8.11 | 0.02 | 1.21 | **0.30** | **0.34** | 0.50 | 3.51 | **0.37** | **0.02** |
| **MetaDrive Average** | 0.40 | 3.21 | 0.25 | 1.45 | -0.06 | 0.24 | 0.63 | 10.26 | 0.13 | 3.18 | **0.36** | **0.25** | 0.33 | 2.33 | **0.54** | **0.10** |

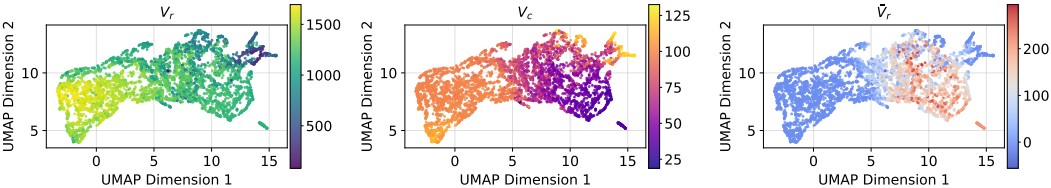

Figure 3: UMAP visualization of $V_r$, $V_c$, and $\overline{V}_r$ across 5000 random states in the *BallRun* dataset. Each dot corresponds to a distinct state.

## 5.2 ABLATION STUDY AND ANALYSIS

Here we present ablation studies on the key components of our method. Additional ablation results are provided in Appendix E.

**Visualization Results.** Figure 3 presents a UMAP projection (McInnes et al., 2020) of the value functions $V_r$, $V_c$, and $\overline{V}_r$ learned on the *BallRun* task, visualized with a subset of $5,000$ randomly sampled states. We observe that states with high $V_r$ values frequently coincide with high $V_c$ values, suggesting that purely maximizing $V_r$ may lead the policy into unsafe states, especially under inaccurate $V_c$ estimates. In contrast, $\overline{V}_r$ assigns low values to states with high $V_c$, while preserving value trends similar to $V_r$ in other regions, demonstrating the ability of $\overline{V}_r$ to jointly optimize for reward and safety during policy learning.

Moreover, we observe that $\overline{V}_r$ assigns nearly identical values to unsafe states, consequently providing insufficient incentive to escape such regions. This observation necessitates a policy designed to actively minimize $V_c$ whenever the agent is in an unsafe state.

**Ablation on Safety Threshold.** We evaluate the sensitivity of our method to the safety threshold

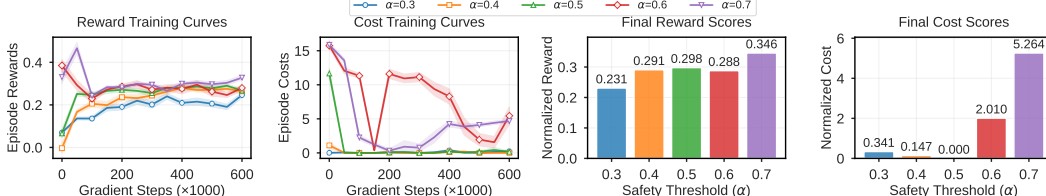

Figure 4: Training curves and normalized scores for different thresholds on the *BallRun* task (Final scores are evaluated over 20 episodes and 3 seeds)

Table 2: Ablation on different design choices. Gray: Unsafe agents. **Bold**: Safe agents whose normalized cost is smaller than 1.

| Task | $w/o\ \overline{V}_r$ | | $w/o\ V_c$ | | *AWR actor* | | Ours | |
|------|-------------|-------------|-------------|-------------|-------------|-------------|-------------|-------------|
| | reward ↑ | cost ↓ | reward ↑ | cost ↓ | reward ↑ | cost ↓ | reward ↑ | cost ↓ |
| BallRun | 0.94 | 14.69 | **0.29** | **0.09** | **0.08** | **0.05** | **0.30** | **0.00** |
| AntRun | 0.62 | 5.73 | 0.61 | 1.16 | **0.63** | **0.43** | **0.62** | **0.53** |
| AntCircle | 0.58 | 13.00 | **0.38** | **0.20** | 0.34 | 3.12 | **0.32** | **0.23** |
| DroneRun | 0.71 | 13.28 | **0.60** | **0.94** | **0.47** | **0.60** | **0.59** | **0.26** |
| DroneCircle | 0.85 | 18.5 | **0.53** | **1.50** | **0.47** | **0.75** | **0.52** | **0.32** |
| CarGoal1 | 0.59 | 1.18 | **0.35** | **0.48** | **0.33** | **0.62** | **0.42** | **0.57** |
| CarGoal2 | 0.36 | 2.41 | **0.30** | **0.37** | **0.20** | **0.32** | **0.32** | **0.67** |

$\ell$ by testing three different values. The safety thresholds are selected as $\alpha$-quantile of $V_c$ values in datasets. The results are presented in Figure 4. Since actions that violate the safety constraint lead to termination in an absorbing state, the choice of $\ell$ directly influences the performance. When $\alpha \leq 0.5$, the normalized score increases steadily with $\alpha$. For $\alpha > 0.5$, however, performance declines and the normalized cost exhibits oscillations during training.

**Ablation on Action Chunk Length.** To validate the efficacy of action chunking length in value function learning, we conducted ablation studies on *BallRun* and *CarGoal2* tasks. As the different time horizon in Safety-Gymnasium and Bullet-Safety-Gym, we choose $h \in \{1, 3, 5, 7\}$ in *BallRun* and $h \in \{1, 5, 10, 15\}$ in *CarGoal2* for our experiment. As shown in Fig. 5, using values of $h > 1$ leads to consistent performance improvements over the baseline. Specifically, the normalized reward increases steadily with the chunk length $h$, while the normalized cost decreases.

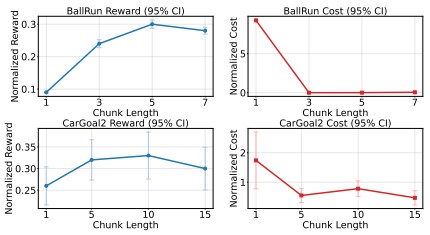

Figure 5: Ablation over chunk length ($h \in \{1, 3, 5, 7\}$) on *BallRun* and ($h \in \{1, 5, 10, 15\}$) on *CarGoal2* tasks.

**Ablation on Key Components** We evaluate the effectiveness of key components in our method, particularly the region-based reward-penalized value function. To assess the contribution of the penalized value function, we compare against two ablated variants: (1) *w/o $\overline{V}_r$*, which uses the vanilla $V_r$ without safety-aware penalization as the objective of safe states, and (2) *w/o $V_c$*, which does not incorporate the cost value function $V_c$ for unsafe states. To further demonstrate that our performance is not overly dependent on a specific policy extraction and parameterization approach, we also include an ablation with a three-layer MLP Gaussian actor trained via AWR, denoted as *AWR actor*.

As summarized in Table 2, the variant *w/o $\overline{V}_r$* exhibits severe constraint violations due to the lack of safety-aware value guidance. *w/o $V_c$* achieves strong constraint satisfaction in most tasks, but fails in a subset of them as a result of pushing the safety threshold $\ell$ to its limit in pursuit of higher reward. The *AWR actor* yields constraint satisfaction comparable to our full method but attains lower reward, likely due to the limited expressivity of the parametric policy. In contrast, our complete approach effectively balances constraint satisfaction and reward maximization.

The consistent safety performance of the latter three variants, all of which include the proposed $\overline{V}_r$ function, underscores the effectiveness of our penalized value formulation. Notably, the full constraint satisfaction achieved by the *AWR actor* across all tasks suggests that the safety assurance of our method is robust to the choice of actor architecture.

## 6 CONCLUSION

In this work, we propose a novel approach that utilizes IQL to learn safe and high-performing policies from offline dataset. We introduce a penalized reward formulation by integrating cost constraints into reward learning, enabling to approximate the optimal value function within a safe policy space. We employ action chunking to enhance the numerical stability of the training process and mitigate the effect of bootstrap errors, thereby improving the robustness of value function estimation. We provide theoretical analysis, which shows that our penalized value function ensures policy safety. Experiments on DSRL benchmarks demonstrate that our method outperforms existing baseline algorithms in both safety and return, achieving the highest returns in 13 out of 17 tasks while fully satisfying safety constraints.

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

# A  THEORETICAL INTERPRETATIONS

## A.1  PROOF TO PROPOSITION 1

*proof.* Let $V^\pi_{-C}$ denotes the cost value function in reward penalty MDP $\mathcal{M}_C$, is defined as

$$V^\pi_{-C}(s) = \mathbb{E}_{s_0=s,a_t\sim\pi,s_t\sim P_{-C}}\Big[\sum_{t=0}^{T-1}\gamma^t\mathbb{I}(s_t=s_a)(-C)\Big], \tag{22}$$

where $P_{-C}$ is the transition dynamic of $\mathcal{M}_C$.

Consider an unsafe policy $\pi_1$ in the original MDP $\mathcal{M}$, such that $V^{\pi_1}_c(\rho_0) \geq \ell$, and assume the cost function $c \in \{0,1\}$. Then in $\mathcal{M}_C$, we have $V^{\pi_1}_{-C}(\rho_0) \leq -C\ell$, since the absorbing nature of $\mathcal{M}_C$ leads to a higher visitation frequency of unsafe states under $\pi_1$ compared to $\mathcal{M}$. Then,

$$\overline{V}^{\pi_1}_r(\rho_0) = \mathbb{E}_{s_0\sim\rho_0,a_t\sim\pi_1,s_t\sim P_{-C}}\Big[\sum_{t=0}^{T-1}\gamma^t\overline{r}_t\Big]$$

$$= \mathbb{E}_{s_0\sim\rho_0,a_t\sim\pi_1,s_t\sim P_{-C}}\Big[\sum_{t=0}^{T-1}\gamma^t(\mathbb{I}(s_t\neq s_a)r_t + \mathbb{I}(s_t=s_a)(-C))\Big]$$

$$= \mathbb{E}_{s_0\sim\rho_0,a_t\sim\pi_1,s_t\sim P_{-C}}\Big[\sum_{t=0}^{T-1}\gamma^t(\mathbb{I}(s_t\neq s_a)r_t\Big]$$

$$\quad + \mathbb{E}_{s_0\sim\rho_0,a_t\sim\pi_1,s_t\sim P_{-C}}\Big[\sum_{t=0}^{T-1}\gamma^t\mathbb{I}(s_t=s_a)(-C))\Big]$$

$$\leq \mathbb{E}_{s_0\sim\rho_0,a_t\sim\pi_1,s_t\sim P}\Big[\sum_{t=0}^{T-1}\gamma^t r_t\Big] + \mathbb{E}_{s_0\sim\rho_0,a_t\sim\pi_1,s_t\sim P_{-C}}\Big[\sum_{t=0}^{T-1}\gamma^t(\mathbb{I}(s_t=s_a)(-C))\Big]$$

$$= V^{\pi_1}_r(\rho_0) + V^{\pi_1}_{-C}(\rho_0)$$

$$\leq V^{\pi_1}_r(\rho_0) - C\ell$$

$$\leq \sum_{t=0}^{T-1}\gamma^t r_{\max} - C\ell. \tag{23}$$

Now, assume that for any $\ell \geq 0$, there exists at least one safe policy $\pi_2$ such that $V^{\pi_2}_c(\rho_0) \leq l$. Setting $\ell = 0$, $\pi_2$ never enters $s_a$, and thus the minimum reward it receives in $\mathcal{M}_c$ is $\sum_{t=0}^{T-1}\gamma^t r_{\min}$. To ensure that $\pi_1$ is suboptimal, it suffices to choose $C$ such that:

$$\sum_{t=0}^{T-1}\gamma^t r_{\max} - C\ell < \sum_{t=0}^{T-1}\gamma^t r_{\min}. \tag{24}$$

Rearranging, we derive

$$C > \frac{\sum_{t=0}^{T-1}\gamma^t(r_{\max}-r_{\min})}{\ell}. \tag{25}$$

For the infinite-horizon case where $T = \infty$, this simplifies to

$$C > \frac{r_{\max}-r_{\min}}{(1-\gamma)\ell}. \tag{26}$$

## A.2  DISCUSSION OF OPTIMALITY

In this section, we will show that when $C \to \infty$, the optimal policy in the penalized MDP $\mathcal{M}_C$ is also the optimal policy in the original MDP $\mathcal{M}$ with $\ell = 0$.

Let $\pi^*_C$ denote the optimal solution to $\mathcal{M}_C$, and $\pi^*$ denote the optimal solution to $\mathcal{M}$. When $l = 0$, $V^{\pi^*}_c(\rho_0) = 0$. Therefore, $\pi^*$ will never enter the unsafe region (i.e., all states with state visitation distribution $d^{\pi^*} > 0$ have zero cost). In $\mathcal{M}_C$, $\pi^*$ will never receive the $-C$ penalty.

As $C \to \infty$, $\pi_C^*$ will also never receive the $-C$ reward, since there exists at least one policy (e.g., $\pi^*$) whose value is greater than $\sum_{t=0}^{T-1} \gamma^t r_{\min}$. Hence, the state visitation distribution of $\pi_C^*$ in $\mathcal{M}_C$ is zero over the absorbing state $s_a$. Excluding $s_a$, the transition structure of $\mathcal{M}_C$ is identical to that of $\mathcal{M}$. Therefore, $\pi_C^*$ is also the optimal policy for $\mathcal{M}$.

A.3 PROOF TO PROPOSITION 2

*proof.* For a MDP $\mathcal{M}$, we define a new MDP $\mathcal{M}_h$. $M_h$ has the same state space $\mathcal{S}$ as $\mathcal{M}$. The new action space $\mathcal{A}_h$, the $h$-step reward function $r_h$, and transition probabilities $P_h$ is:

$$\mathcal{A}_h = \underbrace{\mathcal{A} \times \mathcal{A} \times \cdots \mathcal{A}}_{h \text{ times}}, \tag{27}$$

$$r_h(s_t, a_{t:t+h}) = \sum_{k=0}^{h-1} \gamma^k r_{t+k}, \tag{28}$$

$$P_h(s_{t+h}|s_t, a_t) = \sum_{s_{t+1}, \cdots, s_{t+h-1}} \prod_{k=t}^{t+h-1} P(s_{k+1}|s_k, a_k). \tag{29}$$

Then we consider the offline setting, where data are collected by a behavior policy $\mu(a_{t:t+h}|s_t)$. Following Theorem 3 of Kostrikov et al. (2021), we have

$$\lim_{\tau \to 1} V_\tau(s_t) = \max_{\substack{a_{t:t+h} \in \mathcal{A}_h \\ \text{s.t. } \mu(a_{t:t+h}|s_t) > 0}} Q^*(s_t, a_{t:t+h}). \tag{30}$$

The above discussion says that IQL with action chunking still approximates an optimal value function in $\mathcal{M}_h$. While the optimal action for each state is a chunk of actions in the dataset. We further extend the $h$-step IQL to obtain the explicit form of the corresponding implicit policy. Following Theorem 4.1 of Hansen-Estruch et al. (2023), we write the objective as

$$\arg \min_{V(s_t)} \mathbb{E}_{a_{t:t+h} \sim \mu}[f(Q(s_t, a_{t:t+h}) - V(s_t))],$$

where $f$ is arbitrary convex function, and $f = L^\tau$ in IQL. Note that the objective function is convex with respect to $V(s)$

$$
\begin{aligned}
0 &= \frac{\partial}{\partial V(s)} \mathbb{E}_{a_{t:t+h} \sim \mu}[f(Q(s_t, a_{t:t+h}) - V(s_t))]\Big|_{V=V^*} \\
&= -\mathbb{E}_{a_{t:t+h} \sim \mu}[f'(Q(s_t, a_{t:t+h}) - V^*(s_t))] \\
&= \mathbb{E}_{a_{t:t+h} \sim \mu}\left[\frac{|f'(Q(s_t, a_{t:t+h}) - V^*(s_t))|(Q(s_t, a_{t:t+h}) - V^*(s_t))}{|Q(s_t, a_{t:t+h}) - V^*(s_t)|}\right].
\end{aligned}
\tag{31}
$$

We then define the implicit policy to be

$$\pi_{\text{imp}}(a_{t:t+h}|s_t) = \frac{\mu(a_{t:t+h}|s_t)|f'(Q(s_t, a_{t:t+h}) - V^*(s_t))|}{Z_{\text{imp}}|Q(s_t, a_{t:t+h}) - V^*(s_t)|}, \tag{32}$$

where $Z_{\text{imp}}$ is a normalization constant, and rewrite the above expression as

$$
\begin{aligned}
&= \mathbb{E}_{a_{t:t+h} \sim \pi_{\text{imp}}}[(Q(s_t, a_{t:t+h}) - V^*(s_t))] \\
&= \frac{\partial}{\partial V(s_t)} - \frac{1}{2} \cdot \mathbb{E}_{a \sim \pi_{\text{imp}}}[(Q(s_t, a_{t:t+h}) - V(s_t))^2]\Big|_{V=V^*} = 0.
\end{aligned}
\tag{33}
$$

With the above discussion, we obtain the action chunking policy is also a reweighted form of behavior policy like 1-step IQL do. For loss in Eq. 15, we have that the implicit policy is

$$\pi_{\text{imp}}(a_{t:t+h}|s_t) \propto \frac{\mu(a_{t:t+h}|s_t)|f'(\overline{Q}(s_t, a_{t:t+h}) - \overline{V}^*(s_t))|}{|\overline{Q}(s_t, a_{t:t+h}) - \overline{V}^*(s_t)|}. \tag{34}$$

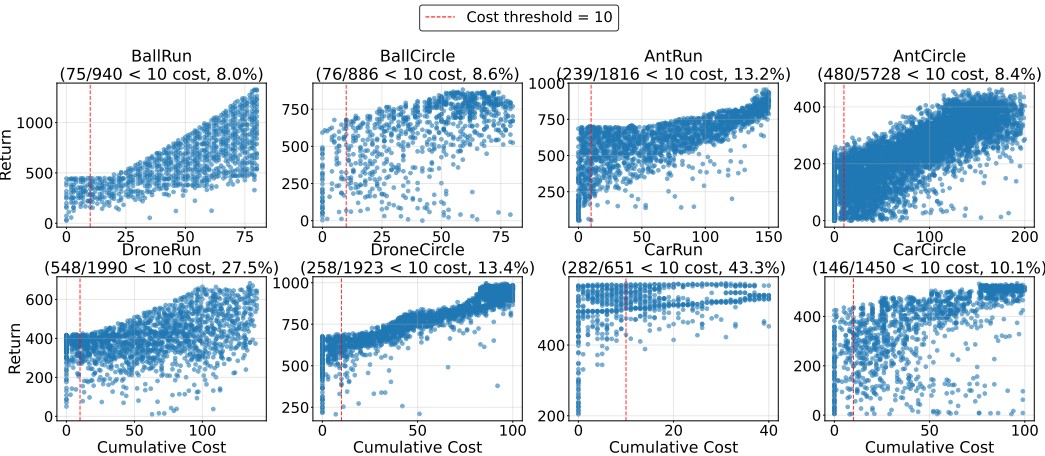

Figure 6: Dataset Composition Overview for Bullet-Safety-Gym Tasks.

# B  BENCHMARK DETAILS

## B.1  SAFETY GYMNASIUM

Safety-Gymnasium (Ray et al., 2019), a MuJoCo-based benchmark for safe RL, evaluates the trade-off between performance and safety through two categories of environments. The first involves obstacle avoidance tasks performed by a *Car* agent across three scenarios (*Goal*, *Button*, and *Push*), each with two difficulty levels (1 and 2). These tasks represent an important class of real-world reinforcement learning problems in which an agent must move through an environment and interact with objects to achieve a goal. The agent receives rewards for task completion and penalties for contacting hazards. These environments are named in the format {*Agent*}{*Task*}{*Difficulty*} (e.g., *Car-Goal1*). The second category consists of velocity-constrained environments featuring three agents (*Ant*, *HalfCheetah*, and *Swimmer*), where the objective is to maximize forward movement reward while adhering to safe speed limits. This tests the ability to balance performance with precise control to avoid failures due to overspeed. Environments in this category follow the naming scheme AgentVel (e.g., *AntVelocity*).

## B.2  BULLET SAFEY GYM

Bullet-Safety-Gym (Gronauer, 2022) is a safe RL benchmark based on the PyBullet engine. Unlike Safety-Gymnasium, it uses shorter time horizons to enable faster training, serving as its efficient complementary alternative, with a broader variety of robotic agents, including *Ball*, *Car*, *Drone*, and *Ant*. The task settings remain relatively straightforward, offering two main types: *Circle* and *Run*. Each environment is named concisely using the convention {*Agent*}{*Task*}, (e.g., *AntCircle* ).

## B.3  COMPOSITION OF THE DATASET

The datasets used in our evaluations predominantly contain a mix of safe and unsafe trajectories. The safety compliance of policies during evaluation is determined by their adherence to the undiscounted cost-return threshold. We provide further details regarding the dataset composition.

Figure 6 and Figure 7 show the reward and cost returns of trajectories in the dataset, with each point representing one pre-collected trajectory. Most datasets used in our evaluation contain both safe and unsafe trajectories. The cost limit is indicated by a red dashed line for reference. A significant number of trajectories exceed this limit, and unsafe trajectories often outnumber safe ones.

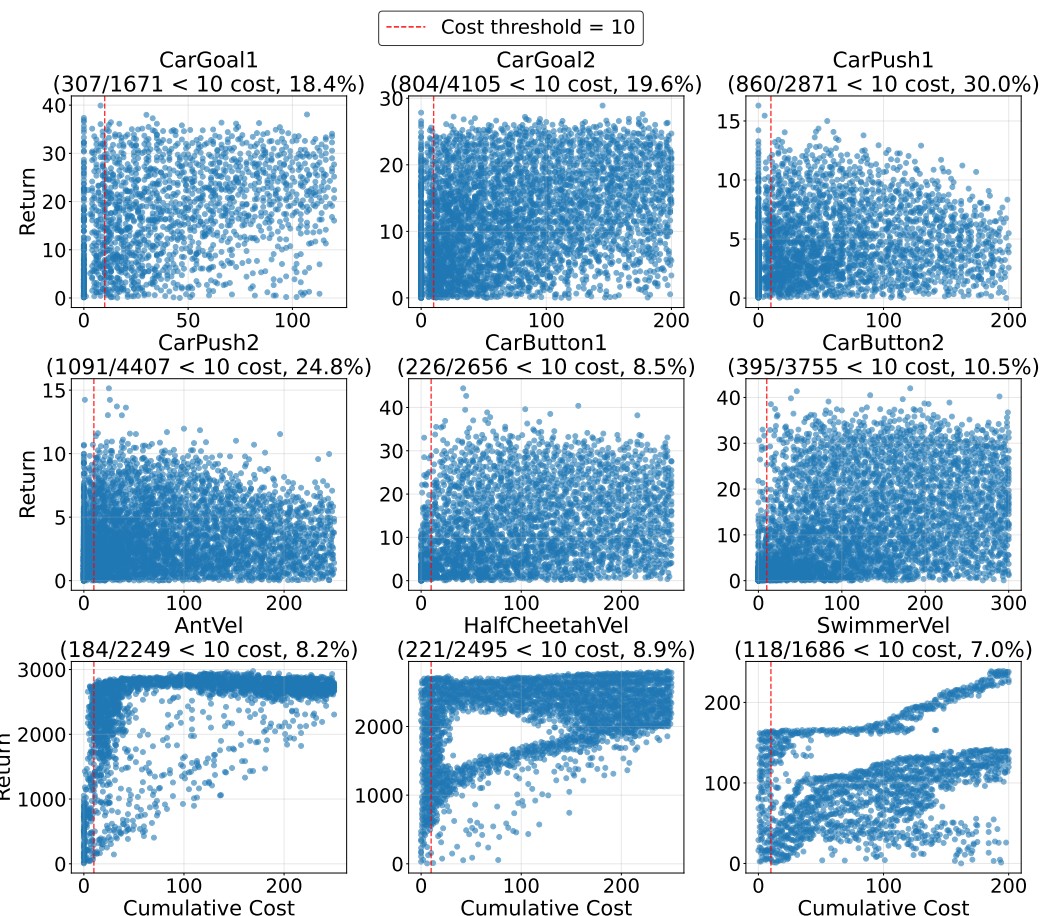

Figure 7: Dataset Composition Overview for Safety-Gymnasium Tasks.

| **Algorithm 1** CFG Training | **Algorithm 2** CFG Sampling |
|---|---|
| **for** Each gradient steps **do** | $a \sim \mathcal{N}(0, I)$ |
| $\quad (s, a) \sim D, \ a_0 \sim \mathcal{N}(0, I), \ t \sim U(0, 1)$ | $t \leftarrow 0$ |
| $\quad$ Label with optimality $o \in \{0, 1\}$. | **for** $n \in [0, \dots, N-1]$ **do** |
| $\quad$ If $\mathrm{rand}() < 0.1$, set optimality $o = \emptyset$. | $\quad v = (1 - w)\, v_\theta(a, t, s, \emptyset) + w\, v_\theta(a, t, s, o = 1)$ |
| $\quad a_t \leftarrow (1 - t)\, a_0 + t\, a$ | $\quad a \leftarrow a + (n/N)v$ |
| $\quad \theta \leftarrow \nabla_\theta \|v_\theta(a_t, t, s, o) - (a - a_0)\|^2$ | $\quad t \leftarrow t + (n/N)$ |
| **end for** | **end for** |
| | **return** $a$ |

## C  Flow Matching and Classifier Free Guidance

**Flow Matching.**  Flow matching is an expressive diffusion-style generative model that enjoys simplicity compared to denoising diffusions. Given a data distribution $p(x) \in \Delta(\mathbb{R}^d)$ on a $d$-dimensional Euclidean space, flow matching aims to construct a time-dependent vector field $v_\theta : [0, 1] \times \mathbb{R}^d \to \mathbb{R}^d$ such that a flow $\phi : [0, 1] \times \mathbb{R}^d \to \mathbb{R}^d$ is described by:

$$\frac{d}{dt}\phi_t(x) = v_\theta(\phi_t(x)), \tag{35}$$

where $\phi_1(x) = x$. A simplest variant of flow matching is based on linear paths and uniform time sampling, and trained by minimizing the following loss:

$$\mathbb{E}_{x^0 \sim \mathcal{N}(0, I_d), x^1 \sim p(x), t \sim \mathrm{Unif}[0,1]}\left[ \|v_\theta(t, x^t) - (x^1 - x^0)\|^2 \right], \tag{36}$$

where $\mathcal{N}(0, I_d)$ is the $d$-dimensional standard normal distribution, $\mathrm{Unif}[0, 1]$ denotes the uniform distribution, and $x^t = (1 - t)x^0 + tx^t$.

**Classifier Free Guidance.**  Classifier-free guidance (CFG) (Ho and Salimans, 2021) uses Bayes' rule to guide the generated distribution to the desired direction. CFG trains a single diffusion model to implement both conditional and unconditional ones. Rather than sampling in the direction of a trained classifier's gradient, the CFG approach derives the guided distribution through the following formula:

$$\nabla_x \log q(x) = \nabla_x \log p(x) + \omega(\nabla_x \log p(x|y) - \nabla_x \log p(x)), \tag{37}$$

where $q(x)$ is the guided distribution.

**Policy Improvement by CFG.**  In recent work, Frans et al. (2025) proposed the use of CFG to steer a diffusion policy toward optimality, demonstrating strong empirical performance. The authors parameterize policies as a product of two components: a reference policy $\hat{\pi}$ and an optimality function $f : \mathbb{R} \to \mathbb{R}$. The policy is defined as follows:

$$\pi(a|s) \propto \hat{\pi}(a|s) \cdot f\big(A(s, a)\big), \tag{38}$$

where $A(s, a)$ denotes the advantage function. It is shown that if $f$ is non-negative and monotonically increasing with respect to $A$, then $\pi$ is guaranteed to improve upon $\hat{\pi}$. This result can be generalized to an exponentiated form:

$$\pi(a|s) \propto \hat{\pi}(a|s) \cdot f\big(A(s, a)\big)^\omega, \tag{39}$$

where $\omega$ controls the degree of improvement.

In practice, the optimality function $f$ is cast as a binary random variable $o \in \{\emptyset, 0, 1\}$, with a likelihood proportional to $f$: $p(o|s, a) \propto f(A(s, a))$. A flow-matching instantiation of this framework is detailed in Algorithm 1 and Algorithm 2.

## D  Implement Details

Our implementation is based on PyTorch (Paszke et al., 2019). All experiments were conducted on a single NVIDIA RTX 4090 GPU, with each run completed within 4 hours. Further implementation details are provided in the following section.

### D.1 PSEUDOCODE

To provide an intuitive understanding of our method, we present a brief summary of it in this subsection.

---

**Algorithm 3** R2PAC

---

1: Initialize parameters $\overline{Q}_r, Q_c, \overline{V}_r, V_c, v_\theta$
2: Value function learning ($h$-step IQL):
3: **for** each gradient steps **do**
4:     get batch of $(s_t, a_{t:t+h}, s_{t+h}) \sim D$
5:     update $\overline{Q}_r$, $Q_c, \overline{V}_r$ and $V_c$ by the Eq. 15 and Eq. 16
6:     update target networks
7: **end for**
8: Policy extraction (CFG):
9: **for** each gradient steps **do**
10:     get batch of $(s_t, a_{t:t+h}) \sim D$
11:     update vector filed $v_\theta$ using Eq. 19
12: **end for**

---

Through the action chunking technique, action sequences are employed as inputs to the Q-function during training. We extend the 1-step approach employed in IQL to incorporate $h$ consecutive actions. Specifically, we utilize transitions of the form $(s_t, a_{t:t+h}, s_{t+h})$, and apply expectile regression, using the same state-value function estimation method as IQL, to approximate the optimal value function. For $\mathcal{L}_{Q_r}$, the loss calculation includes an additional penalty term for safety constraint violations, as defined in Eq. 7.

Our policy extraction is performed using $h$-step value functions. We employ a CFG method for policy extraction, based on Frans et al. (2025), with slight adjustments to the optimality criteria. Our actor is designed to generate a sequence of $h$ consecutive actions. During execution, however, only the first action in each sequence is executed, consistent with standard 1-step actors.

### D.2 COMPUTATIONAL COSTS

We report the training time and GPU memory utilization for our method (R2PAC) and the baselines on the *CarGoal1* task in Table 3. All models were trained for one million gradient steps. With the exception of TREBI, FISOR and CAPS(IQL), all baseline implementations are sourced from the OSRL library (Liu et al., 2024). We re-implemented the TREBI and FISOR algorithm ourselves, while for CAPS(IQL), we used the official code from the authors to obtain the results. All algorithms employed their original hyperparameters as reported in the respective publications.

| | Training time | CUDA memory |
|---|---|---|
| BC | 34min | 580MB |
| CDT | 13h36min | 4466MB |
| CPQ | 2h24min | 638MB |
| COptiDICE | 1h40min | 594MB |
| TREBI | 26h23min | 1016MB |
| FISOR | 2h51min | 728MB |
| CAPS(IQL) | 1h47min | 620MB |
| R2PAC(ours) | 3h20min | 880 MB |

Table 3: Computational costs of baselines and our method on *CarGoal1* task.

### D.3 HYPERPARAMETER

To facilitate understanding and analysis of hyperparameter effects, we categorize them into two classes: unified hyperparameters for all DSRL experiments (Table 4) and per-task hyperparameters (Table 5).

Table 4: Hyperparameters for DSRL experiments.

| Hyperparameter | Value |
|---|---|
| Learning rate | 0.0003 |
| Optimizer | Adam (Kingma and Ba, 2014) |
| Gradient steps | 600000 |
| Minibatch size | 1024 |
| MLP dimensions | $[512, 512, 512, 512]$ |
| Nonlinearity | GELU (Hendrycks and Gimpel, 2023) |
| Flow steps | 32 |
| Flow time sampling distribution | $\text{Unif}([0, 1])$ |
| CFG scale $w$ | 2.0 |
| Generated candidate numbers | 16 |
| Chunk length $h$ | 1 for *HalfCheetahVel* task and 5 for others |
| Expectile $\tau$ | 0.9 |
| Discount factor $\gamma$ | 0.99 |
| Target critic soft update | 0.005 |
| Reward scale | 200.0 for *Car* series task in Safey-Gymnasium, 1.0 for others |
| Cost scale | 1.0 |

To achieve optimal performance across tasks, we carefully tuned our algorithm's key hyperparameters. The hyperparameter configurations are provided in Table 4. The CFG scale $\omega$ balances task performance against behavioral regularization: a higher $\omega$ prioritizes high-value actions, while a lower $\omega$ adheres more closely to the behavior policy. These selections reflect inherent task requirements. In our experiments, we found that a mild guidance scale ($\omega = 2.0$) consistently achieved high performance, and was therefore kept fixed across all tasks.

The cost threshold $\ell$, which serves as a metric to separate safe and unsafe regions, significantly influences performance. A high value of $\ell$ may cause the policy to overlook safety constraints, whereas a low value can overly constrain the policy, hindering reward maximization. Fine-tuning $\ell$ is a challenging and labor-intensive process. To reduce the workload, we propose a principled approach that sets $\ell$ as the $\alpha$-quantile of the dataset value. As illustrated in Figure 6, datasets vary in their composition of safe and unsafe trajectories. Empirically, we observe that datasets with a higher proportion of safe trajectories tolerate a larger $\ell$, leading to improved performance without constraint violation. Conversely, for datasets with fewer safe trajectories, a smaller $\ell$ is preferable. Accordingly, we adaptively set $\ell$ as the $\alpha$-quantile of the state value $V_c$ during training. This approach eliminates the need for manual tuning of $\ell$ and thus yields robust and high-performing results in our experiments.

Another important hyperparameter is the penalty coefficient $\overline{C}$. Theoretically, setting $\overline{C}$ as large as possible is preferred. However, in practice, an excessively large $\overline{C}$ may hinder value function approximation due to limited model capacity. Empirically, we observe that our method is relatively insensitive to the choice of $\overline{C}$. Therefore, we offer only two options for $\overline{C}$, as shown in Table 5, to simplify hyperparameter tuning.

# E ADDITIONAL EXPERIMENTAL RESULTS

## E.1 ADDITIONAL VISUALIZATION RESULTS

We provide more UMAP visualization over $V_r$, $V_c$, and $\overline{V}_r$ to elucidate the advantage of our method in Figure.8.

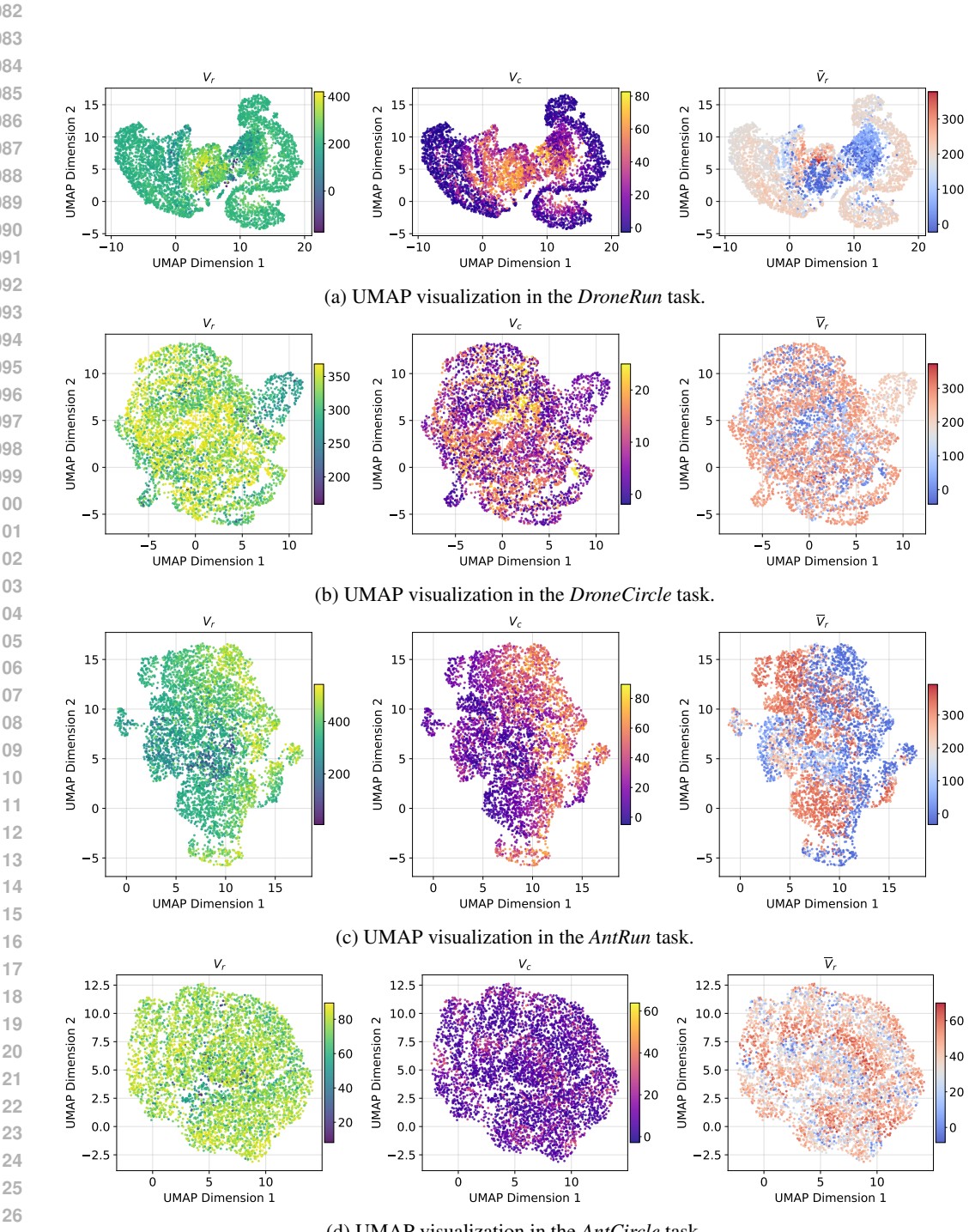

(a) UMAP visualization in the *DroneRun* task.

(b) UMAP visualization in the *DroneCircle* task.

(c) UMAP visualization in the *AntRun* task.

(d) UMAP visualization in the *AntCircle* task.

Figure 8: Comparative visualization of learned state representations using UMAP dimensionality reduction across four distinct locomotion tasks: *DroneRun*, *DroneCircle*, *AntRun*, and *AntCircle*.

Table 5: Per-task hyperparameters selection

| Task | Quantile $\alpha$ | Penalty Coefficient $\overline{C}$ |
|------|-------------------|-------------------------------------|
| AntRun | 0.7 | 1.0 |
| BallRun | 0.5 | 1.0 |
| CarRun | 0.95 | 1.0 |
| DroneRun | 0.7 | 1.0 |
| AntCircle | 0.7 | 1.0 |
| BallCircle | 0.85 | 1.0 |
| CarCircle | 0.9 | 1.0 |
| DroneCircle | 0.7 | 1.0 |
| CarPush1 | 0.9 | 10.0 |
| CarPush2 | 0.9 | 10.0 |
| CarGoal1 | 0.9 | 10.0 |
| CarGoal2 | 0.9 | 10.0 |
| CarButton1 | 0.8 | 10.0 |
| CarButton2 | 0.5 | 10.0 |
| AntVelocity | 0.95 | 1.0 |
| HalfCheetahVelocity | 0.99 | 1.0 |
| SwimmerVelocity | 0.7 | 1.0 |

### E.2 ADDITIONAL ABLATION ON POLICY EXTRACTION CHOICE

We demonstrate that the $h$-step penalized reward value function is the most effective component of our method. To verify the irrelevance of the policy extraction manner, we compare the CFG actor with the AWR actor.

In Section 5, we have conducted the ablation studies using a three-layer MLP Gaussian actor, which is trained by

$$\mathcal{L}_\pi = \mathbb{E}_{(s,a)\sim D}\left[\left(\mathbb{I}_{s\in\mathcal{S}_{\text{safe}}}\exp(\alpha(\overline{Q}_r - \overline{V}_r) + \mathbb{I}_{s\in\mathcal{S}_{\text{safe}}}\exp(\beta(V_c - Q_c)\right)\log\pi_\phi(a|s)\right], \qquad (40)$$

where $\alpha, \beta > 0$ denote the temperature parameter. In our experiment, we set both $\alpha$ and $\beta$ as 6.0.

To isolate the effect of the policy parameterization method, we also do an extra experiment using the flow-matching framework for the AWR actor. We denote it as *AWR-Flow*. The loss function is defined as:

$$\begin{aligned}
\mathcal{L}_v = \mathbb{E}_{(s_t, a_{t:t+h})\sim D}\Big[\Big(&\mathbb{I}_{s\in\mathcal{S}_{\text{safe}}}\exp(\alpha(\overline{Q}_r - \overline{V}_r) \\
&+ \mathbb{I}_{s\in\mathcal{S}_{\text{safe}}}\exp(\beta(V_c - Q_c)\Big)||v_\theta(a_{t:t+h}^i, i, s_t) - (a_{t:t+h} - a_{t:t+h}^0)||^2\Big].
\end{aligned} \qquad (41)$$

As shown in Figure 9, both *AWR-Flow* and our CFG method demonstrate enhanced safety assurance and achieve higher final rewards in most tasks by maximizing $\overline{V}_r$.

### E.3 ADDITIONAL ABLATION ON FLOW POLICY SETTINGS

In the main experiments, we employ 32 flow steps for action generation and consider 16 candidates for rejection sampling, which incurs a non-trivial computational overhead. This configuration is adopted to ensure a fair comparison with prior works that utilize similar settings for diffusion or flow-based policies (Zheng et al., 2024; Hansen-Estruch et al., 2023; Frans et al., 2025; Chen et al., 2023). To verify that our approach does not fundamentally rely on such computationally intensive components, we conduct an ablation study using simplified variants with only 8 flow steps or only a single candidate (i.e., without rejection sampling). As shown in Table 6, the results indicate that our method maintains strong performance even under these reduced settings, demonstrating its robustness and efficiency without dependence on excessive flow steps or rejection sampling mechanisms.

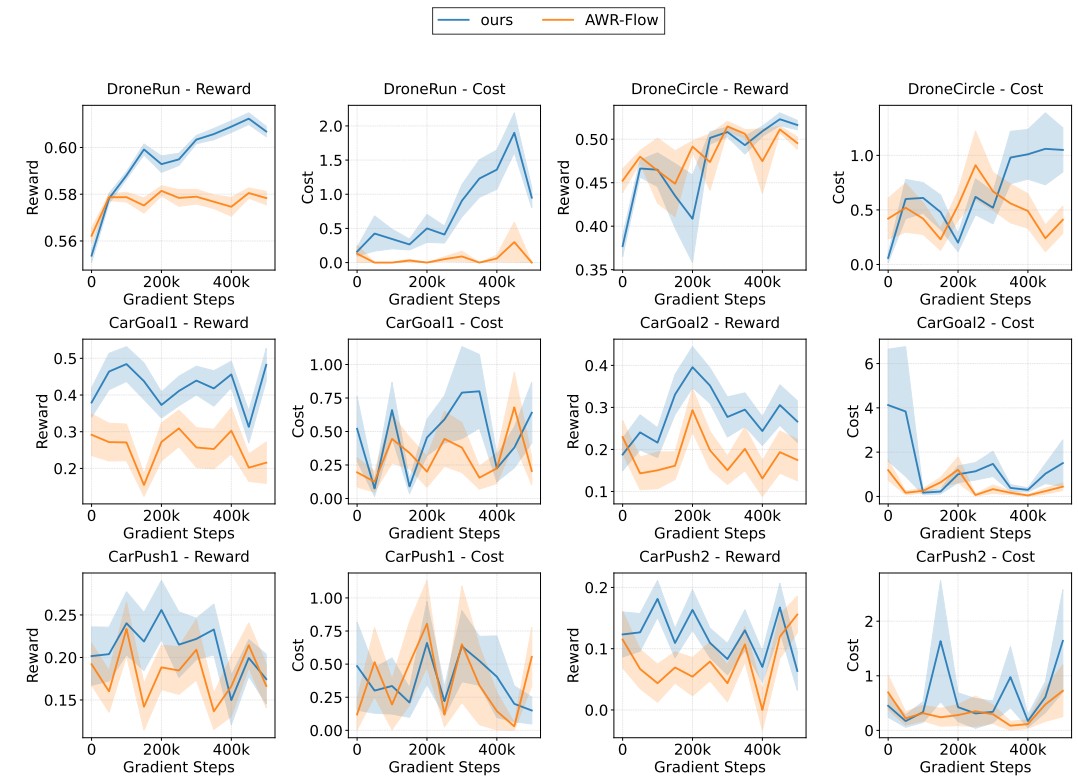

Figure 9: Training curves of our method and *AWR-Flow* on *DroneRun*, *DroneCircle*, *CarGoal1*, *CarGoal2*, *CarPush1*, and *CarPush2* tasks

| Task | 8 steps + 16 candidates | | 32 steps + 1 candidate | | 32 steps + 16 candidates | |
|---|---|---|---|---|---|---|
| | reward ↑ | cost ↓ | reward ↑ | cost ↓ | reward ↑ | cost ↓ |
| PointGoal1 | 0.35 | 0.28 | 0.45 | 0.57 | 0.36 | 0.20 |
| PointGoal2 | 0.31 | 0.27 | 0.42 | 1.01 | 0.48 | 0.95 |
| CarGoal1 | 0.43 | 0.59 | 0.38 | 0.80 | 0.42 | 0.57 |
| CarGoal2 | 0.23 | 0.30 | 0.19 | 0.41 | 0.32 | 0.67 |

Table 6: Ablation study on different flow policy settings.

### E.4 Additional Ablation on Penalty Coefficient

Our method is robust to the choice of the penalty coefficient $\overline{C}$, eliminating the need for extensive hyperparameter tuning. To empirically demonstrate this, we conducted an ablation study varying $\overline{C}$ from 1 to 15 on *CarGoal1* and *CarGoal2* tasks. As shown in Table 7, the performance remains stable across this wide range of values, confirming its insensitivity.

| Task | $\overline{C} = 1$ | | $\overline{C} = 5$ | | $\overline{C} = 10$ | | $\overline{C} = 15$ | |
|---|---|---|---|---|---|---|---|---|
| | reward ↑ | cost ↓ | reward ↑ | cost ↓ | reward ↑ | cost ↓ | reward ↑ | cost ↓ |
| CarGoal1 | 0.41 | 0.78 | 0.43 | 0.79 | 0.42 | 0.57 | 0.45 | 0.26 |
| CarGoal2 | 0.35 | 0.49 | 0.32 | 0.28 | 0.32 | 0.67 | 0.27 | 0.25 |

Table 7: Ablation study on different penalty coefficient.

## E.5 ADDITIONAL ABLATION ON SAFETY THRESHOLD

In R2PAC, we set the threshold $\ell$ as the $\alpha$-quantile of the $V_c$ values in the dataset. While this design introduces additional complexity for manual tuning and lacks a direct interpretable form, it is motivated by a fundamental difference in how the underlying MDP is formulated for value function learning. Specifically, our implementation differs from many baseline methods in a key aspect: the definition of episode termination for value function learning.

For engineering convenience, it is common practice in many methods (Yassine Chemingui, 2025; Xu et al., 2022; Liu et al., 2023b) to treat both timeout (termination due to the maximum step limit) and genuine task termination as the "done" signal. Under this setting, near-infinite horizon environments are effectively truncated into fixed-horizon environments, simplifying value function learning. In such a fixed-horizon formulation, the safety threshold can be approximated using the following formula:

$$\ell = \frac{\kappa(1 - \gamma^T)}{(1 - \gamma)T},\tag{42}$$

where $T$ is the episode length. Although this approach can stabilize training, the learned value function only approximates the value within a modified finite-horizon MDP, not the true value function of the original problem, thereby introducing bias into value estimates. It is worth noting that if these baseline methods were to remove the timeout termination signal, their training would often become highly unstable or even fail to converge.

Unlike the baselines, R2PAC uses only genuine task terminations as the episode ending signal. This design allows us to approximate the true value function of the original task, which is essential for accurate long-horizon safety reasoning.

In this setting, the threshold formula in Eq. 42 is no longer applicable. For a discounted infinite-horizon MDP, the only long-term constraint that can be strictly satisfied in theory is defined by a zero cost threshold; any finite, positive threshold $\ell > 0$ is ill-defined over an infinite trajectory. However, in practice, a cost value of zero is rarely observed due to function approximation limitations and the nature of the data distribution. Directly using $0$ as the threshold would lead to overly conservative policy extraction. Therefore, we adopt a data-driven adaptive approach by setting $\ell$ as the $\alpha$-quantile of the $V_c$ values in the dataset. This is not an arbitrary tuning strategy but a principled substitute in the absence of a closed-form solution.

We emphasize that had we employed the same fixed-horizon MDP setting, our method could also utilize a unified threshold rule. We conducted experiments using both terminal and timeout signals as episode endings, applying the threshold from Eq. 42 instead of the $\alpha$-quantile tuning. The results, shown in Table 8, demonstrate that even under varied episode-ending conditions, our method remains effective.

Although our approach complicates the threshold setting, it offers a significant safety advantage. Using timeout as a termination signal poses a potential safety risk. For instance, states immediately preceding hazardous regions may never be recorded as unsafe in the dataset if episodes consistently end due to the step limit. As a result, these states are erroneously classified as safe, which can lead the learned policy to take hazardous actions during deployment. Our method, by learning a value function that respects the true task horizon, provides a more accurate safety assessment.

| | timeouts+terminals | | terminals only | |
| Task | reward ↑ | cost ↓ | reward ↑ | cost ↓ |
| --- | --- | --- | --- | --- |
| PointGoal1 | 0.38 | 0.28 | 0.36 | 0.20 |
| PointGoal2 | 0.31 | 0.48 | 0.48 | 0.95 |
| CarGoal1 | 0.19 | 0.55 | 0.42 | 0.57 |
| CarGoal2 | 0.37 | 0.49 | 0.32 | 0.67 |

Table 8: Ablation study on different episode termination signals.

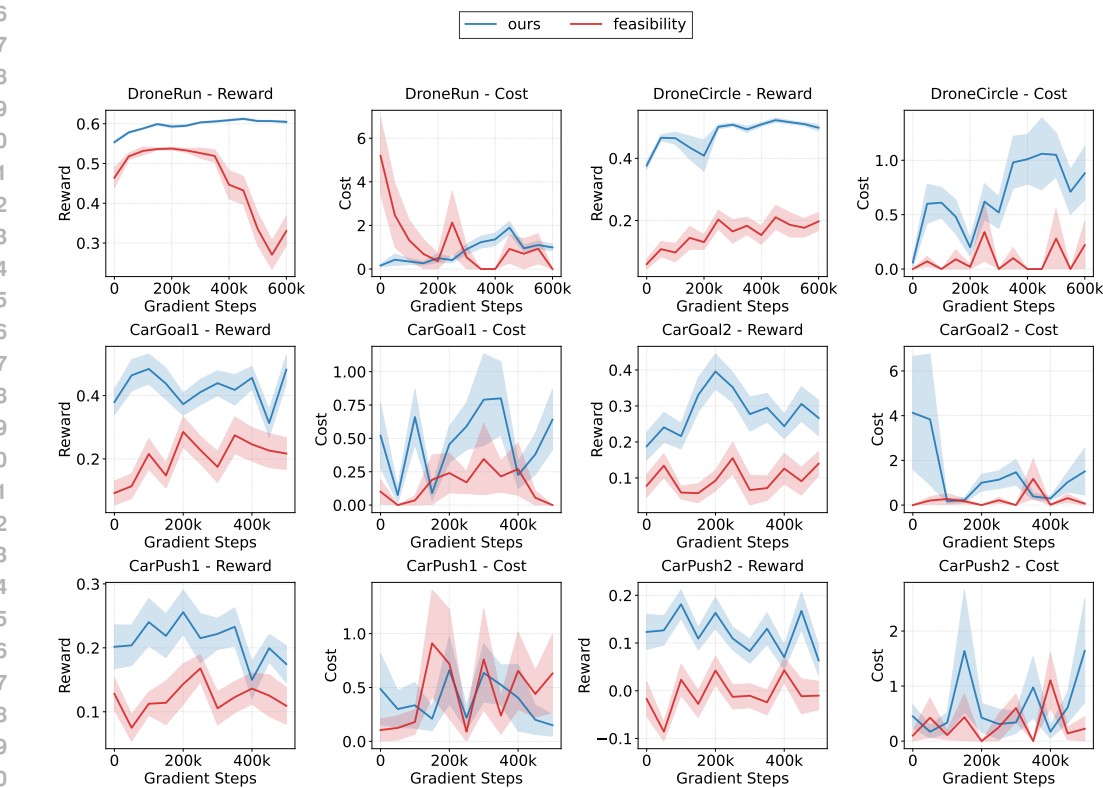

Figure 10: Training curves of our method and *feasibility* on *DroneRun*, *DroneCircle*, *CarGoal1*, *CarGoal2*, *CarPush1*, and *CarPush2* tasks

### E.6 COST METRIC CHOICE

Our method can be readily adapted to different safety metrics. A particularly promising metric called feasibility (Zheng et al., 2024; Yu et al., 2022; Bansal et al., 2017) has recently been introduced in RL, which is Hamilton–Jacobi (HJ) reachability, notable for its hard constraint guarantees and Bellman backup-style updating. The feasibility Q-function is updated as

$$Q_h(s,a) \leftarrow (1-\gamma)h(s) + \gamma \max\{h(s), V_h(s')\}, \tag{43}$$

where $h$ is the state constraint function.

We evaluate a variant of our method, denoted as *feasibility*, wherein HJ reachability is adopted as the criterion for distinguishing safe and unsafe regions while keeping all other components unchanged. As shown in Figure 10, this variant maintains a plausible constraint-satisfaction capability. We note a slight performance drop, which we attribute to the strict, hard-constraint nature of the HJ reachability condition.

## F TRAINING CURVE

We evaluate our method on 17 tasks from the DSRL benchmark. The training curves are presented in Figure 11 and Figure 12. All experiments use the same hyperparameters specified in Table 4 and Table 5. The policy is evaluated for 20 episodes at each checkpoint.

## G LIMITATIONS

One limitation of our current study stems from the need to fine-tune the safety threshold $l$ across different datasets, which is particularly challenging due to varying proportions of unsafe transitions.

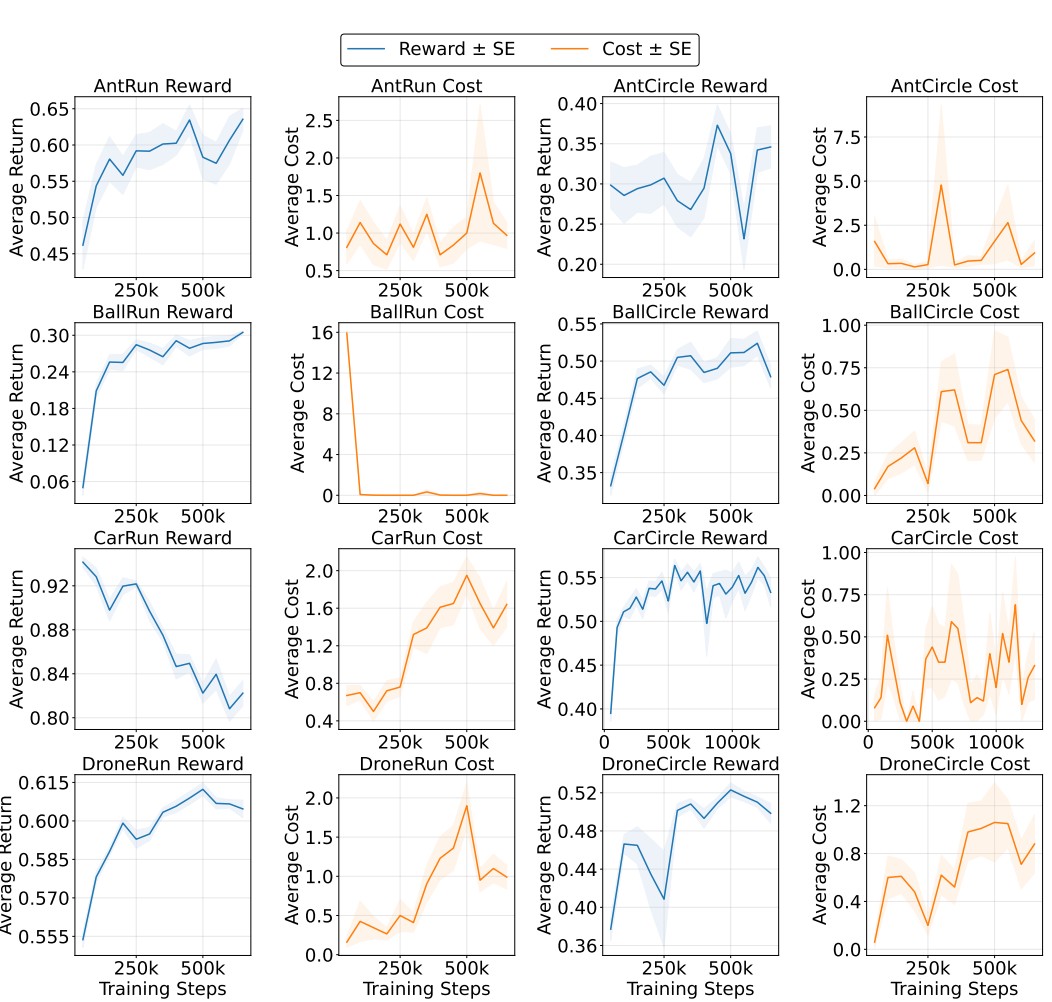

Figure 11: Training curves on the Bullet-Safety-Gym tasks.

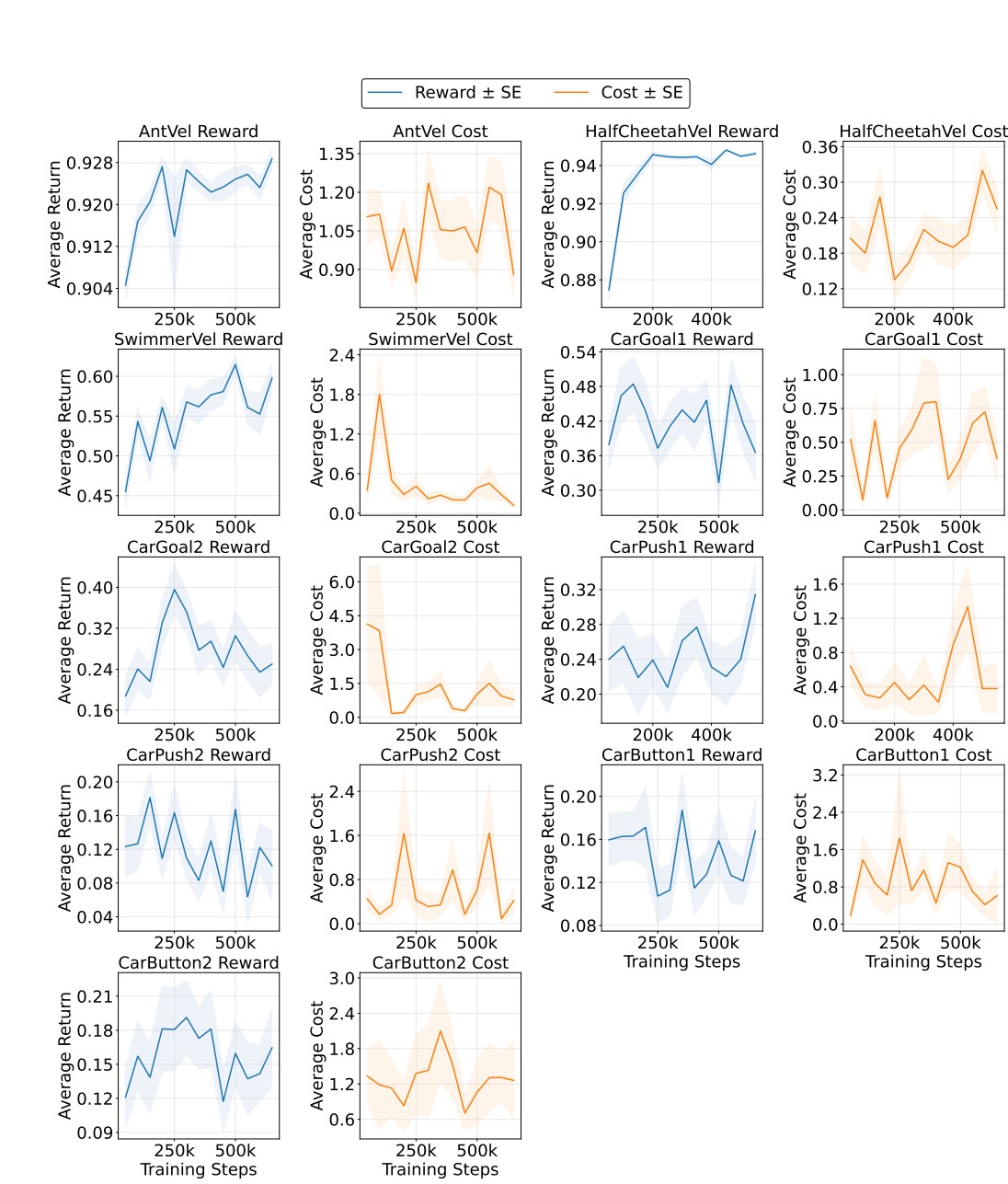

Figure 12: Training curves on the Safety-Gymnasium tasks.

To mitigate this issue, we introduce a simple adaptive thresholding method based on the $\alpha$-quantile. Looking forward, we plan to develop more resilient cost metrics that are robust to uncertain or unsafe data distributions, such as feasibility analysis.

