# OpenReview forum: "Balancing Safety and Return: Region-based Reward Penalty over Action Chunks For Offline Safe RL"
_ICLR.cc/2026/Conference — ICLR 2026 Conference Withdrawn Submission_

### Official Review · Reviewer_uDmP · 2025-10-17

**Soundness:** 3
**Presentation:** 3
**Contribution:** 2
**Rating:** 4
**Confidence:** 5

**Summary:**

This paper proposes R2PAC, an offline safe reinforcement learning method based on reward penalty and action chunking. R2PAC updates the reward–action value function using a reward penalty, introduces action chunks to improve the accuracy of value estimation, and finally performs policy updates through conditioned flow matching. The method achieves strong reward performance on the DSRL benchmark.

**Strengths:**

- The paper is clear and well-written.
- Applying action chunking in offline safe RL to improve value estimation accuracy and enable implicit prediction of future outcomes is a reasonable and effective approach.
- The paper provides a comprehensive set of ablation studies.

**Weaknesses:**

- The proposed approach appears to be a combination of several existing methods, including the reward penalty from Safe MBPO, action chunking, the expectile value learning used in IQL and FISOR, and FISOR’s safe/unsafe region-specific policy update objectives together with its generative model–based policy extraction. As a result, the novelty of the contribution is somewhat limited, and the work feels more incremental in nature. That said, the application of action chunking in this context is, in my view, a notable and valuable contribution.
- There is some ambiguity between the hard-constraint and soft-constraint settings. The overall framework seems intended to address hard constraints, yet the reward penalty introduces a safety threshold hyperparameter $\mathcal{l}$ that is determined by the quantile of dataset values, rather than being a fixed threshold manually specified. This design choice appears inconsistent with the motivation.
    - If the goal is to handle hard constraints, $\mathcal{l}$ should ideally be set to zero.
    - If the setting is not hard-constrained, then $\mathcal{l}$ should instead correspond to the externally defined safety threshold κ.

    Although introducing a tunable hyperparameter may improve empirical performance, it reduces the conceptual clarity and persuasiveness of the method’s formulation.

- During policy extraction, the method employs conditioned flow matching. It is not entirely clear why flow matching was chosen over diffusion-based approaches. Furthermore, the paper replaces FISOR’s AWR with a conditioning mechanism whose design seems somewhat strange: it only distinguishes whether the reward and cost advantages in safe and unsafe regions are ≥ 0 or ≤ 0. However, if multiple actions satisfy these conditions, how are better actions selected among them? Is this achieved solely through rejection sampling based on cost values?
- I have some concerns about the evaluation criteria. If the paper is positioned as addressing a hard constraint setting, then safety satisfaction should be considered more important than reward maximization—that is, one should minimize cost first and then maximize reward under that constraint. However, the paper seems to focus primarily on reward maximization when cost < κ. Based on the results shown in the appendix, if the objective were to minimize cost, the flow AWR variant would actually perform better than the proposed conditioned version.
- The authors should include more experiments on Safety Gymnasium Navigation tasks (e.g., PointXXX environments) to provide a more comprehensive evaluation of the proposed method’s generality and robustness.

**Questions:**

See weaknesses.

---

> ### Author Response · Authors · 2025-11-21
> **Response to Reviewer uDmP (Part 1/2)**
>
> **Weakness 1: Limited novelty of the contribution.**
>
> We sincerely thank the reviewer for their thoughtful feedback and for recognizing the value of applying action chunking in this context. R2PAC is not a simple combination of existing components; rather, it introduces a novel and systematic framework for value function learning. While individual techniques like reward penalty or action chunking exist, their roles in our framework are novel and critical.
>
> The region-based reward penalty is not just an application but a theoretical mechanism to unify the objective. It allows us to learn a single penalized value function $\overline{V}_r$ that inherently represents the optimal value within the safe policy set (Prop. 1, Eq. 12-13). This is a conceptual leap from prior methods like FISOR that often leads to conservatism.
>
> Further, we are the first to identify and leverage action chunking not just for policy robustness, but as a critical stabilizer for value function learning. Action chunking directly mitigates value learning instability by reducing the frequency of these error-prone updates (Sec. 4.2, Fig. 2).
>
> Moreover, the policy is derived primarily from the unified $\overline{V}_r$. Unlike FISOR, the use of $V_c$ in our method is merely a practical, non-core safeguard against value function approximation errors. The policy is inherently safe even without it, as shown in Sec. 5.2.
>
> **Weakness 2: Ambiguity between the hard-constraint and soft-constraint settings and design choice appears inconsistent with the motivation.**
>
> We thank the reviewer for this insightful comment. Our framework is designed for hard constraints, and the use of the dataset $\alpha$-quantile for the threshold $\ell$ is a practical, robust implementation choice that induced by different underlying MDP formulation.
>
> Our method is rooted in the theory of CMDPs for hard constraints. As proven in Proposition 1 and Appendix A.2, with a sufficiently large penalty C, the optimal policy under our penalized value function $\overline{V}_r$ is guaranteed to satisfy the safety constraint $V_c^\pi(\rho_0) \leq \ell$ with zero violation in expectation. This is a hard-constraint guarantee.
>
> Why $\alpha$-Quantile: Unlike baseline algorithms that often treat timeouts as terminal states for Q-learning, we strictly differentiate between genuine termination and timeouts as the episode ending signal. This allows us to approximate the true Q-function, which is critical for accurate long-term safety assessment. However, in a discounted infinite-horizon CMDP, the only long-term constraint that can be strictly satisfied in theory is a zero-cost threshold, which is practically unattainable due to function approximation limits and dataset properties. In the absence of a closed-form threshold for the true Q-function, we adopt the $\alpha$-quantile of the dataset's Vc values as a principled, data-driven substitute. We have added more experiments on different episode termination scenarios in the Appendix E.5 of the updated manuscript. When episode termination arises from both terminal states and timeouts, the corresponding safety threshold is given by average approximation: $\ell=\frac{\kappa(1-\gamma^T)}{(1-\gamma)T}$, where $T$ is the maximum episode length. The results demonstrate that our method is robust to these variations.
> | Task      | timeouts+terminals           | terminals only |
> |-----------|------------|-----------|
> |           | reward  (cost)  | reward  (cost)  |
> | PointGoal1 | 0.38  (0.28)      | 0.36   (0.20)      |
> | PointGoal2 | 0.31   (0.48)      | 0.48    (0.95)      |
> | CarGoal1   | 0.19    (0.55)      | 0.42    (0.57)      |
> | CarGoal2   | 0.37    (0.49)      | 0.32    (0.67)      |

---

> > ### Author Response · Authors · 2025-11-21
> > **Response to Reviewer uDmP (Part 2/2)**
> >
> > **Weakness 3: Policy extraction methodology and action selection.**
> >
> > We thank the reviewer for these insightful questions. Regarding these concerns, we provide the following responses.
> >
> > 1. Choice of Flow Matching over Diffusion:
> > Flow Matching (FM) and Denoising Diffusion Probabilistic Models (DDPM) are closely related generative frameworks. Our choice of FM was primarily motivated by its training simplicity and computational efficiency. This choice is orthogonal to our core contribution of learning a unified value function.
> >
> > 2. Binary Conditioning of Advantage: In IQL-style learning with $\tau\to1$, the value function $V$ approximates $\max_a Q(s,a)$. Therefore, the advantage $A = Q-V$ is, in theory, non-positive, with $A=0$ only for optimal actions. The binary condition ($A\geq0$ for optimal, $<$ 0 otherwise) is a natural and robust binarization of this theoretical property, accounting for function approximation errors. It instructs the CFG to push the policy towards the set of near-optimal actions.
> >
> >
> > 3. How are better actions selected: The guided flow policy inherently selects actions by sampling from the probability distribution of high-value actions learned from the dataset, not via rejection sampling. This naturally favors more probable and robust actions. Regarding the use of rejection sampling, we adopted this technique only to ensure a fair comparison with prior work, as it is a standard component in many recent diffusion/flow-based policies [1-3]. We have added ablation studies on this mechanism in the Appendix E.3. The results demonstrate that the effectiveness of our method does not rely on the rejection sampling mechanism.
> >
> > **Weakness 4: Concerns about the evaluation criteria.**
> >
> > We thank the reviewer for raising this important point. We fully agree that safety is the primary concern in hard-constrained settings. However, we emphasize that a policy minimizing cost alone is practically valueless, as it fails to accomplish the task objective (e.g., an agent remaining stationary).
> >
> > Our method is designed to explicitly address the true goal of constrained RL: to maximize task performance while satisfying a strict safety constraints. In line with this, our evaluation protocol treats safety as the primary criterion: an algorithm is considered viable only if its normalized cost remains below 1. Among these safe agents, we then compare normalized rewards to identify the best-performing policy under the constraint.
> >
> > Regarding the Flow AWR variant noted by the reviewer: while it may achieve marginally lower cost in certain cases, it does so at the expense of significantly lower reward. This indicates its inability to effectively balance the constrained optimization objective.
> >
> > **Weakness 5: More experiments on Safety Gymnasium Navigation tasks.**
> >
> > We thank the reviewer for the suggestion. We have now included additional experiments on the PointXXX navigation tasks from the Safety-Gymnasium benchmark, as requested. Please refer to Table 1 in the updated manuscript. The results demonstrate that our method consistently satisfies the safety constraints while achieving competitive returns, further validating its generality and robustness across different agent embodiments.
> >
> > [1] Hansen-Estruch, Philippe, et al. "IDQL: Implicit Q-Learning as an Actor-Critic Method with Diffusion Policies." arXiv preprint arXiv:2304.10573, 2023.
> >
> > [2] Huayu, Chen, et al. "Offline Reinforcement Learning via High-Fidelity Generative Behavior Modeling." The Eleventh International Conference on Learning Representations, 2023.
> >
> > [3] Zhendong, Wang, et al. "Diffusion Policies as an Expressive Policy Class for Offline Reinforcement Learning." The Eleventh International Conference on Learning Representations, 2023.

---

> > > ### Comment · Reviewer_uDmP · 2025-11-25
> > >
> > > Thank you very much for the authors’ detailed responses. These replies have indeed addressed some of my concerns, but several concerns still remain.
> > >
> > > 1. **Novelty** — I accept that using action chunking in safe RL for both policy robustness and value function stabilization is a meaningful contribution. However, concerns about novelty remain.
> > >     1. First, the claim that “learning a single penalized value function … is a conceptual leap from prior methods” is overstated. Safe MBPO also adopts a similar reward-penalty framework and similarly requires only a single penalized value function, with comparable theoretical justification.
> > >     2. Second, if the reward-penalty framework is viewed as the core contribution, I do not see clear innovation or necessity in the additional modifications combined with it. Expectile regression for identifying the largest safe region was already introduced in FISOR, and the current work mainly applies it. Action chunking is introduced to enhance policy robustness and stabilize value learning, but its presence or absence does not affect the theoretical properties of reward penalty. Finally, conditioned flow matching as a policy class may improve expressiveness, but it likewise does not influence the theoretical guarantees of reward penalty. Therefore, taken together, these components are effective engineering techniques that improve performance, but their novelty appears incremental.
> > > 2. **Motivation behind** $\mathcal{l}$ — I understand that the authors choose a non-zero $\mathcal{l}$ to reduce the conservativeness of the policy and obtain higher returns at the expense of some safety, which is consistent with the experimental results. However, in a strict hard-constraint setting, such a modification lacks theoretical justification. Although Appendix A.2 provides an optimality analysis, it relies on the limiting case $C\to\infty$, which differs significantly from practical implementation. The role of $\mathcal{l}$ is to identify the largest feasible set, and setting $\mathcal{l}>0$ effectively includes some unsafe states into this feasible region. In fact, from an experimental standpoint, one could similarly introduce a positive l\mathcal{l}l in FISOR to enlarge the feasible region and achieve higher returns at the cost of increased violations. Thus, while I acknowledge the engineering usefulness of this trick in settings where safety constraints are not strict and policies appear overly conservative, such an approach is theoretically unjustified under rigorous hard-constraint requirements.
> > > 3. **Evaluation criteria** — I agree that a policy minimizing cost alone is of little practical value. However, the main issue is: how do we determine the safety boundary? If we set a fixed safety boundary, say 10, then why not simply model the problem using soft-constraint methods, since soft constraints are explicitly designed for fixed, non-zero safety budgets? Otherwise, if the safety boundary is unspecified, the differing performance of policies should be interpreted merely as occupying different points along a Pareto frontier, making it unclear which policy should be considered superior.
> > > 4. **Experiments** — Introducing tunable parameters such as $\alpha$ and $\bar{C}$ while comparing against baselines like FISOR is unfair unless hyperparameters are also adjusted correspondingly for the baselines. If hyperparameters differ across tasks for the proposed method, then baseline hyperparameters should also be allowed similar flexibility. Otherwise, $\alpha$ and $\bar{C}$ˉ should remain fixed across all tasks.
> > >
> > > Taking all the responses into account, I am inclined to keep my score.

---

> > > > ### Author Response · Authors · 2025-11-26
> > > > **Response to Reviewer uDmP (Part 1/2)**
> > > >
> > > > We thank the reviewer for their thoughtful engagement with our work and for the opportunity to further clarify our approach. Please see our response to your concerns below.
> > > >
> > > > **Concern 1: Novelty**
> > > >
> > > > We thank the reviewer for their thoughtful engagement with our work and for the opportunity to further clarify the novelty of our approach.
> > > >
> > > > The reviewer correctly notes that individual components of our method have precedents in the literature. However, we respectfully argue that the core novelty lies in their synergistic integration to address a fundamental challenge in offline safe in-sample learning: the sub-optimality caused by the divergence between the learned policy and the value functions.
> > > >
> > > > Our work is the first to introduce and theoretically justify a region-based reward penalty within the offline, in-sample IQL framework. In offline RL, a central challenge is effective policy regularization. The field is largely divided into two paradigms: conservative methods and in-sample learning (exemplified by IQL). Conservative methods often fail on difficult tasks， while in-sample learning methods are more high-performing and widely adopted in adjacent area. However, a key bottleneck in in-sample learning, especially critical in safe offline RL, is the divergence between the learned policy and the value functions, a problem that is severely compounded when two competing value functions (reward and cost) are learned separately. Please refer to our response to Reviewer KUfP [Weakness 4] for a more detailed discussion. To resolve this, we introduce a novel reward-penalty mechanism that unifies the two objectives into a single value function, thereby directly mitigating the sub-optimality arising from value conflicts and policy-value divergence.
> > > >
> > > > While prior methods such as Safe MBPO do employ a form of reward penalty, they operate in a fundamentally different setting. First, Safe MBPO is an online algorithm, where unlimited environment interaction alleviates the policy-value divergence problem. Second, it relies on the assumption of deterministic dynamics, which is often unrealistic in practical RL settings. Third, its ability to avoid constraint violations is contingent on the accuracy of a learned dynamics model, rather than being driven solely by the reward-penalty mechanism.
> > > >
> > > > In contrast, our motivation for adopting reward penalty is distinct. Our method is designed for offline setting, effective under stochastic dynamics, and independent of any learned dynamic model. Therefore, the sole commonality between our approach and Safe MBPO is the use of reward penalty, which is a general and classic technique, not an innovation unique to Safe MBPO. Our contribution lies in reformulating and deploying this classic idea within a novel, theoretically grounded framework to address a specific and previously unresolved problem in offline safe RL.
> > > >
> > > > In conclusion, while components like action-chunking serve as effective implementation techniques, the core conceptual contribution of our work, a novel framework that resolves the sub-optimality from policy-value divergence, stands as a significant contribution in itself. This principled solution provides the foundation for building reliable and high-performing safe agents, irrespective of the specific auxiliary techniques used for stabilization or policy extraction.

---

> > > > > ### Author Response · Authors · 2025-11-26
> > > > > **Response to Reviewer uDmP (Part 2/2)**
> > > > >
> > > > > **Concern 2: Motivation behind $\ell$**
> > > > >
> > > > > We thank the reviewer for this insightful comment. We would like to address this in three key parts.
> > > > >
> > > > > 1. The theoretical justification for a non-zero $\ell$ is rooted in the fundamental quality and coverage of offline datasets. A threshold of $\ell=0$ is only theoretically sound when the dataset contains a sufficient density of states with exactly zero cost value, which is often not the case in practice. In many real-world scenarios, collected trajectories incur some non-zero cost, making $V_c>0$ for all states in the dataset. For instance, in infinite-horizon tasks such as those in the Safety-Gymnasium benchmark, there are no strictly safe states under a stochastic behavior policy. Given sufficient data coverage, virtually every state will, with probability one, be part of a trajectory that eventually incurs a cost, as the stochasticity ensures that unsafe events occur almost surely over an infinite horizon. Therefore, enforcing $\ell=0$ would result in an empty safe policy set. Instead, the selection of a positive value for $\ell$ is based on the assumption that a subset of states in the dataset is safe. This serves as a necessary adaptation to define a practically useful safe region from an imperfect dataset.
> > > > >
> > > > > 2. In any environment with stochastic dynamics or a stochastic policy, it is not possible to theoretically guarantee absolute safety (i.e., zero cost) with probability 1 over an infinite horizon. Even in online settings, unless all unsafe states are completely unreachable from the safe region, which is impossible, a stochastic policy will yield a non-zero state visitation frequency on some unsafe states. This results in a non-zero cost value $V_c$ for those supposedly "safe" starting states, thereby violating the theoretical definition that safe states must have zero $V_c$. The best one can achieve is to minimize the probability or magnitude of constraint violations. Therefore, insisting on $\ell=0$ is not just practically difficult but theoretically ill-posed for general stochastic CMDPs. Our use of a positive $\ell$ acknowledges this reality and seeks to find the best policy within a practically achievable safety bound.
> > > > >
> > > > >
> > > > > 3. Threshold tuning is a universal challenge, not unique to our method. For instance, FISOR also relies on an analogous hyperparameter to differentiate safe and unsafe states. As detailed in its Appendix D.3, FISOR introduces a pivotal hyperparameter M which enables 0 as the boundary to separate safe and unsafe states. In practice, this M acts as a human-chosen hinge in its feasibility formulation, directly shaping the resulting safe set. This demonstrates that the core challenge of calibrating a safety threshold is a universal issue across different methodological frameworks.
> > > > >
> > > > >
> > > > > **Concern 3: Evaluation criteria**
> > > > >
> > > > >
> > > > > We thank the reviewer for raising this fundamental question.
> > > > >
> > > > > 1. *How do we determine the safety boundary?*
> > > > > As established previously, our objective is to identify the optimal policy within a practically achievable safety bound. While perfect hard constraints are often difficult to satisfy, pursuing agents with high performance and minor violations remains a critical objective, acknowledging the inherent limitations of offline learning. Given the nascent stage of research in this area, we adopted the experimental setting from FISOR, e.g. a cost threshold of 10, to ensure a direct and fair comparison against the current state-of-the-art.
> > > > >
> > > > > 2. *Why not soft-constraint formulation?*
> > > > > While our method is compatible with a soft-constraint formulation, as detailed in Appendix E.5, we primarily do not adopt it because we argue that the common practice of using a fixed, non-zero safety budget is conceptually problematic. As detailed in Appendix E.5 of our updated manuscript, such a budget is fundamentally tied to a finite-horizon setting. However, most budget-based methods in practice arbitrarily truncate the inherent infinite-horizon MDP into a finite one. This lacks rigorous theoretical justification and is primarily a pragmatic compromise to make the problem tractable.
> > > > >
> > > > > **Concern 4: Experiments**
> > > > >
> > > > >
> > > > > We thank the reviewer for this insightful comment.
> > > > >
> > > > > We would like to emphasize that the core strength of our method does not depend on extensive hyperparameter search. As shown in our response to Weakness 2, the parameter $\alpha$ is derived from our problem formulation. More importantly, we have already included ablation studies in the Appendix E.4 and E.5, which demonstrates the robustness of our method's performance to a fixed safety budget and a range of C values across different tasks.
> > > > >
> > > > > Therefore, we confirm that our method's superior performance over the baselines, as presented in the paper, is valid and not an artifact of unfair hyperparameter tuning.

---

### Official Review · Reviewer_kBfL · 2025-10-30

**Soundness:** 2
**Presentation:** 3
**Contribution:** 2
**Rating:** 4
**Confidence:** 3

**Summary:**

This paper proposes R2PAC (Region-Based Reward Penalty over Action Chunks), an offline safe RL method that integrates cost constraints directly into reward learning to improve safety and stability. By penalizing rewards over action chunks that may lead to unsafe transitions, R2PAC enables multi-step value learning within the safe policy space. Experiments on the DSRL benchmark show that it achieves higher returns and lower costs than state-of-the-art methods.

**Strengths:**

The authors provide theoretical analysis showing that the proposed reward penalty framework preserves policy optimality and ensures convergence guarantees.

**Weaknesses:**

The identification of safe and unsafe sets relies on the learned value functions (Eq.(17)), which may be inaccurate during training. As a result, the training process appears unstable, with both reward and cost exhibiting noticeable oscillations in the reported figures (Figures 9, 10, 11).

**Questions:**

1. What are the specific assumptions required for the theoretical propositions (e.g., Propositions 1 and 2) to hold? Do they rely on perfect estimation of $V_c$ and $V_r$, accurate separation of safe and unsafe regions, or bounded approximation errors?

2. In Eq. (4), could the authors clarify why the $Q$-function takes only $s_t$ and the action chunk as inputs? Since $V(s_{t+h})$ depends on the future state $s_{t+h}$, if $Q$ were instead conditioned on both $s_t$ and $s_{t+h}$, how would the corresponding value function be defined? Does the proposed formulation imply that the value function also depends on both states?

3. In the ablation study, the authors mention that “satisfactory evaluation results can still be achieved using model checkpoints” (lines 418–419). Could the authors elaborate on what this statement means in practice? Specifically, are the reported results based on the best-performing checkpoint during training, and how is fairness ensured in such comparisons? For the main results, are the evaluations conducted using the best checkpoint or the final model parameters?

---

> ### Author Response · Authors · 2025-11-21
> **Response to Reviewer kBfL**
>
> **Weakness 1: Training process appears unstable.**
>
> We thank the reviewer for this insightful observation. It is true that the initial inaccuracy of the learned value functions is indeed a primary source of the observed oscillations. Our experimental results align with the anticipation of the reviewer. Once $V_c$ begins to stabilize, it provides a more reliable signal for the reward penalty in $\overline{V}_r$'s learning target. This, in turn, leads to the convergence of $\overline{V}_r$. Therefore, the initial oscillations represent a transient phase of mutual alignment between the value functions, after which both stabilize.
>
> **Question 1: Assumptions required for the theoretical propositions.**
>
> We thank the reviewer for this crucial question regarding the theoretical assumptions.
>
> Regarding the proposition 1, our current theoretical analysis relies on the assumption of a near-perfect estimation of the cost value function $V_c$ to precisely separate the safe and unsafe regions.
>
> We acknowledge that a complete theoretical framework with bounded approximation errors for $V_c$ is an important direction for future work. However, we would like to highlight the Empirical Robustness: Although our theory assumes an accurate $V_c$, our extensive experiments demonstrate that the method is remarkably robust in practice. Even with the inevitable approximation errors from training on a finite offline dataset, our algorithm consistently satisfies safety constraints across all 32 diverse tasks (Table 1 in the updated revision). This indicates that the method may do not require perfect $V_c$ estimation to be effective.
>
> Regarding Proposition 2, it is important to note that its assumptions are not additional or exotic but are in fact the standard and minimal requirements for the IQL paradigm to recover the optimal policy in an offline setting.
>
> **Question 2: Inputs to the Value Functions.**
>
> We thank the reviewer for this insightful question regarding the inputs to the Q-function. We believe the question likely pertains to the chunked Q-function defined in Eq. (14), not Eq. (4). We have updated Eq. (14) in the manuscript for greater clarity and precision. If we have misinterpreted your question, please kindly correct us.
>
> The future state $s_{t+h}$ is $\textit{not}$ an input to the chunked Q-function. Rather, the dependence on the future state is handled through an expectation over the environment's transition dynamics. This formulation does not imply that the Q-function is conditioned on $s_{t+h}$.
>
> **Question 3: Model checkpoint selection and evaluation fairness.**
>
> For our main results (Table 1) and all subsequent analyses, reported scores are based on evaluations using the final model parameters obtained after training completion.
>
> The original statement, "satisfactory evaluation results can still be achieved using model checkpoints," referred to our use of the best-performing checkpoint for generating the bar charts in Figure 4. This approach was taken because, for certain values of $\alpha$, the final model parameters occasionally exhibited high or oscillating cost values. To accurately isolate and demonstrate the underlying performance trend, we evaluated the best checkpoint from each training run, selected based on validation performance.
>
> We acknowledge that this mixed protocol in the ablation could lead to an unfair comparison. Therefore, in response to the reviewer's feedback, we have updated Figure 4. The revised figure now reports results exclusively using the final model parameters, without any checkpoint selection.

---

> > ### Comment · Reviewer_kBfL · 2025-11-24
> >
> > Thank you for the clarifications. However, my main concern about stability remains. The training figures show that the algorithm oscillates not only at the beginning of training but throughout, and more importantly, it oscillates between safe and unsafe policies. In addition, although Figure 2 suggests that the Q-values converge faster with larger chunk lengths, it is not clear whether they actually converge to the true values.

---

> > > ### Author Response · Authors · 2025-11-26
> > > **Response to Reviewer kBfL**
> > >
> > > We thank the reviewer for the follow-up comment and the opportunity to further clarify the stability properties of R2PAC. We agree that training stability is crucial, especially in safety-critical applications. Below we provide additional evidence and analysis to address the reviewer’s concerns:
> > >
> > > 1. The observed oscillations during training do not indicate that the training process is unstable. The curves in Figures 9-12 reflect intermediate policy performance evaluated with only 20 episodes per checkpoint. With such a small sample size, the evaluated cost is subject to high statistical variance. In contrast, the final evaluation is more reliable as it is conducted over 20 episodes across 3 random seeds. Furthermore, cost oscillations during training are a common phenomenon observed in many related offline safe RL methods. For instance, similar oscillatory patterns are evident in Figure 11 of FISOR [1] and Figure 5b of LSPC [2], even though these methods employ a simpler value learning framework without the additional complexity of our coupled mechanism.
> > >
> > > 2. The convergence of the proposed $\overline{V}_r$ objective is theoretically grounded. The cost value function $V_c$ is inherently convergent, as it is trained using the standard IQL framework. Once $V_c$ stabilizes, it provides a consistent signal for the reward penalty, enabling $\overline{V}_r$ to converge through its Bellman updates. The convergence rate, while difficult to quantify precisely, is a general challenge in value-based RL and not a specific limitation of our method.
> > >
> > > 3. We would like to clarify a potential misinterpretation: the observed training oscillations are not an inherent flaw of our $\overline{V}_r$ learning objective. Instead, the evidence from our ablation studies (Appendices E.2 and E.6) strongly suggests that the instability originates from other components, such as the cost value function $V_c$ and the Classifier-Free Guidance (CFG) policy extraction method. Results from these experiments show that alternative design choices lead to more stable training with reduced oscillation. This directly implicates the $V_c$ function and the CFG sampling process as the primary sources of oscillation. It is critical to distinguish these standard, "off-the-shelf" components from our novel contribution, which itself learns a stable and coherent value function. Furthermore, we emphasize that CFG was employed solely as a conventional and high-performing tool for extracting policies from diffusion/flow models. Its use is orthogonal to the core contribution and theoretical soundness of our $\overline{V}_r$ framework. The instability introduced by these auxiliary components is a known, general challenge in the field, not a specific failure of our proposed method.
> > >
> > > 4. It is important to note that policy oscillations between safe and unsafe policies occur only in a small subset of the most challenging tasks in the benchmark, such as CarPush2 and CarButton2. These specific environments are widely recognized as difficult cases where a significant number of offline safe RL methods fail to learn even a feasible policy.
> > >
> > > 5. Despite transient oscillations during training, the final policies are consistently safe and high-performing. As shown in Table 1, R2PAC achieves costs below the safety threshold in all tasks and attains the highest reward in most of them. This demonstrates that the training process reliably converges to safe and near-optimal policies.
> > >
> > >
> > > Regarding Figure 2, we would like to clarify that the chunked values are guaranteed to converge to the true optimal values in the MDP formulation. The theoretical justification, condensed due to space constraints, is as follows.
> > >
> > > Let's denote the optimal value in the induced MDP as $V_h$ and the optimal value in the original MDP as $V$. $V$ satisfies the Bellman equation: $V(s_t)=\max_{a_t} r(s_t,a_t)+\gamma \mathbb{E}V(s_{t+1})$. By recursively expanding this over a horizon h, we obtain: $V(s_t)=\max_{a_{t:t+h}}\sum_{k=0}^{h-1}\gamma^k r(s_{t+k},a_{t+k})+\gamma^h \mathbb{E}V(s_{t+h})$. The optimal value function $V_h$ in the induced h-step MDP satisfies the Bellman equation: $V_h(s_t)=\max_{a_{t:t+h}}\sum_{k=0}^{h-1}\gamma^k r(s_{t+k},a_{t+k})+\gamma^h \mathbb{E}V_h(s_{t+h})$.
> > >
> > > Note that both $V$ and $V_h$ satisfy the same functional equation. Since the Bellman operator is a contraction mapping, the fixed point of this equation is unique. Therefore, $V_h$ must converge to $V$, implying that the chunked value learning process recovers the true optimal value function.
> > >
> > >
> > >
> > > [1] Yinan, Zheng, et al. "Safe Offline Reinforcement Learning with Feasibility-Guided Diffusion Model." ICLR 2024.
> > >
> > > [2] Prajwal, Koirala, et al. "Latent Safety-Constrained Policy Approach For Safe Offline Reinforcement Learning." ICLR 2025.

---

### Official Review · Reviewer_vzEV · 2025-10-31

**Soundness:** 3
**Presentation:** 3
**Contribution:** 3
**Rating:** 6
**Confidence:** 3

**Summary:**

R2PAC addresses a key limitation in offline safe RL: separately trained reward and cost value functions produce conflicting policies. The method trains a single h-step value function within the safe policy space by penalizing rewards for unsafe action chunks. Key contributions: (1) region-based reward penalty integrating cost constraints, (2) action chunking for stability and temporal consistency, (3) flow-matching policy extraction. Results: highest returns in 13/17 DSRL tasks while maintaining safety across all tasks.

**Strengths:**

- Figure 1 effectively illustrates how decoupled value functions lead to suboptimal trajectories. The problem is real and well-articulated.

- First to integrate action chunking into in-sample OSRL. Region-based penalty (Eq. 7, 12-13) elegantly incorporates safety into reward learning. Proposition 1 provides theoretical guarantees.

- Normalized cost < 1.0 in all 17 tasks, highest rewards in 13 tasks. Substantially outperforms FISOR and other baselines.

**Weaknesses:**

- Safety threshold ℓ (α-quantile) varies from 0.5-0.99 across tasks (Table 4)
- Proposition 1 assumes binary costs but experiments use continuous costs
- Only two C values (1.0, 10.0) used; claims "insensitivity" without evidence
- Flow-matching with 32 steps + 16 candidates likely expensive

**Questions:**

1. How sensitive is performance to h > 15? What's the relationship to task horizon?

2. Computational cost vs baselines (time, memory)?

3. Why does C differ by 10× between task families? How to select for new tasks?

4. Figure 3 shows V assigns similar values to unsafe states—doesn't this contradict sufficiency claims and necessitate Eq. 17?

5. How do V_c estimation errors affect safe region identification and final safety?

---

> ### Author Response · Authors · 2025-11-21
> **Response to Reviewer vzEV (Part 1/3)**
>
> **Weakness 1: Safety threshold $\ell$ ($\alpha$-quantile) varies from 0.5-0.99 across tasks (Table 4).**
>
> We thank the reviewer for this observation. The variation in the safety threshold $\ell$ stems from a fundamental difference in our implementation: the definition of episode termination for Q-function learning.
>
> For engineering convenience, common practice in baseline methods treats both timeout (termination due to the maximum step limit) and genuine task termination as the "done" signal. While this can stabilize learning, it means the learned Q-function approximates the value within a modified finite-horizon MDP, not the true Q-function of the original problem. In contrast, our implementation treats only genuine task termination signals as episode endings. We have added a detailed explanation of this design choice in Appendix E.5.
>
> Therefore, setting the safety threshold $\ell$ as the $\alpha$-quantile of the dataset's $V_c$ values is a practical choice aligned with our implementation. Notably, if we adopt the same episode termination setting as the baselines, $\ell$ can be directly derived from the cost limit $\kappa$ using the relation $\ell = \frac{\kappa(1-\gamma^T)}{(1-\gamma)T}$, where $T$ denotes the episode length. This alternative formulation does not compromise the effectiveness of R2PAC. For further details, please refer to Appendix E.5 of the updated manuscript.
>
> **Weakness 2: Proposition 1 assumes binary costs but experiments use continuous costs.**
>
> We thank the reviewer for this insightful observation.
>
> We would like to clarify that the cost functions in our experimental environments, such as Safety-Gymnasium and Bullet-Safety-Gym, are fundamentally binary in nature. A cost is incurred, for instance, upon collision with a hazard or exceeding a velocity limit. The cost signals are indeed {0, 1}.
>
> **Weakness 3: Only two C values (1.0, 10.0) used; claims "insensitivity" without evidence.**
>
> We thank the reviewer for raising this point. We apologize for the lack of explicit evidence for our claim of "insensitivity" in the initial submission.
>
> To address this directly, we conducted additional ablation studies testing a wider range of values. The results, summarized in the table below, confirm that the performance of R2PAC is indeed robust to the exact value of C. Our choice of 1.0 and 10.0 was simply to cover different scales of reward values in the Safety-Gymnasium and Bullet-Safety-Gym suites. We will include these results in the revised manuscript.
> | Task    | $\overline{C}=1$ |           | $\overline{C}=5$ |           | $\overline{C}=10$ |           | $\overline{C}=15$ |           |
> |---------|------------------|-----------|------------------|-----------|-------------------|-----------|-------------------|-----------|
> |         | reward ↑         | cost ↓    | reward ↑         | cost ↓    | reward ↑          | cost ↓    | reward ↑          | cost ↓    |
> | CarGoal1| 0.41             | 0.78      | 0.43             | 0.79      | 0.42              | 0.57      | 0.31              | 0.35      |
> | CarGoal2| 0.35             | 0.49      | 0.32             | 0.28      | 0.32              | 0.67      | 0.16              | 0.25      |
>
>
> **Weakness 4: Flow-matching with 32 steps + 16 candidates likely expensive.**
>
> We thank the reviewer for the insightful comment. We would like to clarify that our method remains effective even with significantly fewer flow steps and generated candidates. Our primary rationale for this setup (32 steps + 16 candidates) was to ensure a fair and direct comparison with state-of-the-art methods like FISOR.
>
> We conducted an ablation study significantly reducing the number of flow steps and candidates. The results, summarized below, show that performance remains strong even with a substantially more lightweight setup.
> | Task      | 8 steps + 16 candidates |   32 steps + 1 candidate |     32 steps + 16 candidates |
> |-----------|-------------------------|-----------------|-----------------------|
> |           | reward (cost)    | reward (cost)    | reward(cost )    |
> | PointGoal1| 0.35  (0.28)      | 0.45 (0.57)      | 0.36  (0.20)      |
> | PointGoal2| 0.31   ( 0.27 )     | 0.42   (1.01)      | 0.48  (0.95)      |
> | CarGoal1  | 0.43    (0.59)      | 0.38   (0.80)      | 0.42   (0.57)      |
> | CarGoal2  | 0.23    (0.30)      | 0.19   (0.41)      | 0.32   (0.67)      |

---

> > ### Author Response · Authors · 2025-11-21
> > **Response to Reviewer vzEV (Part 2/3)**
> >
> > **Question 1: How sensitive is performance to h $>$ 15? What's the relationship to task horizon?**
> >
> > We thank the reviewer for this insightful question.
> >
> > The reviewer correctly hypothesizes that the optimal chunk length $h$ is indeed related to the task horizon. Through extensive experimentation, we observed that tasks with longer effective horizons generally benefit from a larger $h$. For instance, in the DroneCircle task which has a horizon of 300, we found $h=5$ to be highly effective. Conversely, for the CarGoal2 task with a longer horizon of 1000, a larger value of $h=10$ yielded the best performance.
> >
> > To systematically investigate the sensitivity for $h>15$, we conducted additional experiments on the long-horizon CarGoal2 task, testing values up to $h=25$. The results, summarized in the table below, reveal a pattern of diminishing returns. Performance, measured by normalized reward, improves significantly as $h$ increases from 1 to 10. Beyond $h=15$, the performance gains plateau. This is likely because over-extended chunks, while reducing TD error propagation, begin to suffer from the inherent difficulty of accurately predicting the long-term consequences of action sequences in an offline setting.
> > |       | h=1  | h=5  | h=10 | h=15 | h=20 | h=25 |
> > |-------|------|------|------|------|------|------|
> > | reward | 0.26 | 0.32 | 0.33 | 0.30 | 0.15 | 0.16 |
> > | cost   | 1.74 | 0.55 | 1.47 | 0.47 | 0.78 | 1.05 |
> >
> >
> > **Question 2: Computational cost vs baselines (time, memory)?**
> >
> > We record the training time and GPU memory utilization of our method and baselines in CarGoal1 task in the following table. All models were trained for one million gradient steps.
> > | Method       | Training time | CUDA memory |
> > |--------------|---------------|-------------|
> > | BC           | 34min         | 580MB       |
> > | CDT          | 13h36min      | 4466MB      |
> > | CPQ          | 2h24min       | 638MB       |
> > | COptiDICE    | 1h40min       | 594MB       |
> > | TREBI        | 26h23min      | 1016MB      |
> > | FISOR        | 2h51min       | 728MB       |
> > | CAPS(IQL)    | 1h47min       | 620MB       |
> > | R2PAC(ours)  | 3h20min       | 880MB       |
> >
> > **Question 3: Why does C differ by 10× between task families? How to select for new tasks?**
> >
> > We thank the reviewer for this important question. The primary reason for the 10X difference in C values (1.0 vs. 10.0) between task families is directly tied to the inherent scale of the reward functions in those environments.
> >
> > In the Bullet-Safety-Gym and agent velocity tasks, the native reward magnitudes are relatively small. Therefore, a moderate penalty of C=1.0 is sufficient to deter constraint violations. In the Safety-Gymnasium navigation tasks, the native reward for reaching the goal is also small. To facilitate stable value learning, it is a common practice to scale this reward by a large factor. Consequently, to remain effective within this scaled reward landscape, the penalty must be correspondingly larger, which is why we use C=10.0 for these tasks.
> >
> > Guidance for New Tasks: For a new task, we recommend the following simple procedure: Identify or define a reasonable scale for the reward function. Select a C value that is on the same order of magnitude as the maximum expected cumulative reward. The goal is to ensure that the penalty for entering an unsafe state negates any potential benefit.

---

> > > ### Author Response · Authors · 2025-11-21
> > > **Response to Reviewer vzEV (Part 3/3)**
> > >
> > > **Question 4: Figure 3 shows V assigns similar values to unsafe states—doesn't this contradict sufficiency claims and necessitate Eq. 17?**
> > >
> > > We thank the reviewer for this insightful observation. The behavior shown in the figure does not contradict our sufficiency claims but rather provides the motivation for the practical enhancement in Eq. (17).
> > >
> > > Our core claim is that the penalized value function $\overline{V}_r$ is sufficient to guide the agent to avoid entering the unsafe region from any safe state. However, as the reviewer astutely notes and Figure 3 visualizes, $\overline{V}_r$ assigns similarly low values to states already inside the unsafe region. This is expected and consistent with our framework: $\overline{V}_r$ is trained to maximize reward while never entering unsafe states, so it is not optimized for providing a gradient for escaping them.
> > >
> > >
> > > This is precisely why we introduce the decoupled objective in Eq. (17) as a robust fallback mechanism. It is a practical safeguard against the inherent limitations of function approximation, not a core component of the learning framework. The policy is primarily derived from $\overline{V}_r$, and $V_c$ is only activated as a "rescue policy" in the rare event the agent finds itself in an unsafe state. This design separates the primary objective of safe, high-reward operation from the secondary objective of robust error recovery, which we consider a strength of the overall system.
> > >
> > >
> > > **Question 5: How do $V_c$ estimation errors affect safe region identification and final safety?**
> > >
> > > We thank the reviewer for raising this critical point. A thorough theoretical analysis of this effect is an important area for future work. However, our empirical results provide strong evidence for the robustness of our method to such errors in practice.
> > >
> > > The penalized value function $\overline{V}_r$ is trained to be optimal within the estimated safe region. As shown in Figures 9-11, the policy maintains the normalized cost below 1.0 throughout training, even during early training periods. This means that minor misclassifications at the boundary due to estimation error are often overcome by the learned conservatism of the $\overline{V}_r$ itself.

---

### Official Review · Reviewer_KUfP · 2025-11-01

**Soundness:** 2
**Presentation:** 2
**Contribution:** 2
**Rating:** 2
**Confidence:** 4

**Summary:**

The paper tackles offline safe reinforcement learning’s tendency to violate constraints and the mismatch that arises when reward and cost critics are trained separately, and proposes R2PAC, which embeds safety directly into reward learning via a region-based reward penalty and uses in-sample (IQL-style) learning.  It converts any transition that would enter an unsafe region into an absorbing state with reward −C. To stabilize and speed up training, R2PAC introduces action chunking, h-step critics that reduce bootstrap-error propagation and improve numerical stability compared to 1-step IQL. On the DSRL benchmark, the method reports highest returns on 13 of 17 tasks while keeping costs below strict thresholds on all tasks.

**Strengths:**

- The method demonstrates strong empirical performance, reportedly achieving the highest returns in 13 out of 17 DSRL benchmark tasks while successfully satisfying safety constraints in all of them.
- It identifies and addresses the OSRL problem of the separate training of reward and cost value functions, which can lead to conflicting policy recommendations and suboptimal or unsafe behavior.
- The integration of action chunking and flow-matching enhances the method's performance.

**Weaknesses:**

* The paper's central claim of using a "single value function" is contradictory. It criticizes separating reward and cost functions but then explicitly uses a decoupled objective that relies on both its new penalized reward value,  $\bar{V_{r}}$, and the standard cost value, $V_{c}$ for policy extraction and recovery from unsafe states.

* The R2PAC method relies on per-task tuning of the safety threshold $l$. While acknowledged as a limitation, it makes the achieved performance gains suspect to the tuning. Fisor, for instance, doesn’t rely on such an extensive tuning.

* The paper's reliance on setting the critical safety threshold $l$ as an arbitrary $\alpha$-quantile of the cost values is flawed because this threshold is not shown to have any relationship with the task's true, fixed cost limit (e.g., 5 or 10). If the goal were purely to find the maximally safe policy, the threshold should ideally be set toward zero rather than tuned based on a data percentile.

* The paper's experimental evaluation is lacking as it omits comparisons to many relevant and recent methods.  Please check missing  related work below. A relevant baseline is CAPS. The Figure 1 motivates R2PAC approach by showing a failure mode where myopically selecting the highest-reward, non-unsafe action leads the agent toward a suboptimal, complicated trajectory. CAPS seems to do that by selecting the reward-maximizing action from that safe set at each step.


* The paper's novelty appears incremental. Its core logic (maximize reward in safe states, minimize cost in unsafe states) and components like rejection sampling are conceptually very similar to FISOR, suggesting a "reworked FISOR" that swaps components.

* The claim of "simplicity" for policy extraction is contradicted by the use of a complex flow-matching model (CFG).

* The justification for action chunking is weak. The plots show it changes $Q$-value scales and convergence speed, but this does not prove it produces more *accurate* value estimates or is essential for safety, especially since FISOR performs well without it (in terms of safety).

* The paper appears to copy baseline results directly from the FISOR paper rather than re-running them.

* The abstract's claim that the method is a "drop-in replacement within existing offline RL pipelines" is misleading. The paper focuses almost exclusively on the in-sample learning paradigm and only demonstrates its method as a deep modification of the IQL algorithm. The method is highly integrated with one specific algorithm (IQL) and not shown to be easily applicable to other offline RL methods.

- “we employ rejection sampling to select the action with lowest $Q_c$,” but that tells you if the action is safe by following the safest policy afterwards. It is not the $Q_c$  of the current policy.
- While the authors used the results from the FISOR paper for baselines, they omitted results for the MetaDrive benchmark.
- The literature review also appears incomplete, omitting several relevant OSRL publications, including:
  - CAPS: Constraint-Adaptive Policy Switching [https://www.arxiv.org/pdf/2412.18946]
  - OASIS [https://arxiv.org/pdf/2407.14653]
  - Latent Safety-Constrained Policies [https://arxiv.org/pdf/2412.08794]
  - Trajectory Classification for Safe RL [https://arxiv.org/pdf/2412.15429?]
  -  Constraint-conditioned actor-critic for OSRL [https://openreview.net/pdf?id=nrRkAAAufl]
  - A Similar relabeling idea for the online case. Safe Exploration in Reinforcement Learning.  [https://arxiv.org/pdf/2310.03225]

**Questions:**

- Please check weaknesses.
- "although satisfactory evaluation results can still be achieved using model checkpoints." what does this mean?
- Figure 4 : Why do the "Cost Training Curves" not align with the "Final Cost Scores" bar chart for BallRun? For instance, the $\alpha=0.4$ curve shows costs near 0, but the bar reports a normalized cost of 0.22. Conversely, the $\alpha=0.6$ curve oscillates up to 5, but the bar reports a final normalized cost of only 0.1.
- Figure 10: Is the cost axis in the training curves in Figure 10 (e.g., CarGoal1) showing absolute cost or normalized cost? If the curve shows values between 0.0 and 1.0, how does this reconcile with the normalized cost of 0.57 reported in Table 1, which implies an absolute cost of 5.7?

---

> ### Author Response · Authors · 2025-11-21
> **Response to Reviewer KUfP (Part 1/4)**
>
> **Weakness 1: Clarification on the "single value function" claim.**
>
> We thank the reviewer for the valuable feedback. There is no contradiction in our framework. The objective in Eq. (17) is structured such that the policy is primarily learned and extracted using only $\overline{V}_r$, which directly solves the suboptimality issue illustrated in Fig. 1. The cost function $V_c$ acts purely as a fallback mechanism for robustness during deployment and is not necessary for the core learning process. This is empirically validated in our ablation study (“w/o $V_c$” in Table 2), where policies derived solely from $\overline{V}_r$ maintain strong safety and performance across most tasks.
>
>
> **Weakness 2: The R2PAC method relies on per-task tuning.**
>
> We thank the reviewer for raising this point. The variation in the safety threshold stems from a principled design choice in our underlying MDP formulation, rather than an arbitrary tuning advantage. Our implementation diverges from many baseline methods in a critical aspect: the definition of episode termination for Q-function learning.
>
> Unlike baseline methods including FISOR, which heavily rely on treating timeouts as terminal states to stabilize training and enable the use of a unified safety threshold, our method learns the Q-function solely from genuine task terminations. This design allows us to approximate the true value function of the original task, which is essential for accurate long-horizon safety reasoning. In contrast, the common use of timeouts effectively reduces the problem to a fixed-horizon MDP, thereby biasing value estimates. It is worth noting that if these baseline methods were to remove the timeout termination signal, their training would often become highly unstable or even fail to converge.
>
> This foundational difference directly necessitates different thresholding strategies. In a fixed-horizon MDP, a unified threshold like $\ell=\kappa(1-\gamma^T)/((1-\gamma)T)$ can be derived. In our true infinite-horizon setting, no such closed-form solution exists for a positive $\ell$. Setting $\ell$ via the $\alpha$-quantile of the dataset's $V_c$ values is a principled, data-driven substitute to enforce a stringent safety standard in practice. Had we adopted the same fixed-horizon setting as baselines, our method could also use a unified threshold and would likely achieve comparable or superior performance. The experimental results are presented below and can also be found in the Appendix E.5 of our updated manuscript.
>
> Our approach, while more complex in threshold setting, provides significant safety benefits: It avoids the critical false-negative safety issue caused by timeouts, where states near hazardous regions are incorrectly labeled as safe simply because the episode ended due to steps. It aims to learn the correct, unbiased value function, which we believe is essential for robust and safe long-term deployment.
>
>
> **Weakness 3: Setting the critical safety threshold $\ell$ as an arbitrary $\alpha$-quantile of the cost values is flawed.**
>
> We thank the reviewer for this insightful comment. For an infinite-horizon Q-function, the only threshold with a rigorous, unambiguous interpretation for long-term safety is $\ell=0$. However, due to environment stochasticity and policy stochasticity, it is virtually impossible for a practical value function to satisfy $V_c=0$ exactly. Therefore, enforcing $\ell=0$ is not a feasible objective.
>
> Crucially, any other fixed positive threshold $\ell=0$ also lacks a clear theoretical justification in the infinite-horizon setting and does not guarantee safety. Therefore, setting $\ell$ as an $\alpha$-quantile of the dataset's Vc is not an arbitrary choice, but the most principled and practical implementation available.

---

> ### Author Response · Authors · 2025-11-21
> **Response to Reviewer KUfP (Part 2/4)**
>
> **Weakness 4: Lack of experimental comparisons.**
>
> We thank for the reviewer's suggestion regarding baseline comparisons. We would like to highlight that our original experimental design already included a comprehensive set of baselines representing major methodological directions in offline safe RL: conservative Q-learning (CPQ), sequence modeling (CDT, TREBI), and hard-constraint methods (FISOR). We have now further strengthened this evaluation by including CAPS as an additional representative baseline in our updated manuscript (Table 1).
>
> Upon analysis, we posit that the approach exemplified by CAPS and other methods are, in fact, a representative instance of the fundamental limitation illustrated in Figure 1, and our method is designed specifically to address this. Methods like CAPS rely on independently trained reward ($V_r$) and cost ($V_c$) value functions, following a myopic, projection-based policy: at each step, they first identify a locally safe action set using $V_c$, then myopically select the highest-reward action from it. This greedy, step-wise optimization fails to find a coherent long-horizon policy. The executed trajectory is a reactive compromise at each step, which does not align with the true optimal policy and often leads to suboptimal global paths, as shown in Fig. 1.
>
> In contrast, our R2PAC method makes a fundamental shift: it directly learns a single, unified value function $\overline{V}_r$ that intrinsically encodes long-horizon safety via our region-based penalty. This allows our policy to natively optimize for global reward maximization within the safe policy space, avoiding the myopic pitfalls of projection-based methods. Therefore, CAPS does not resolve the core issue but falls into the trap we identify, while our work provides a principled advancement beyond this paradigm.
>
> **Weakness 5: The paper's novelty appears incremental.**
>
>
> We thank the reviewer for this feedback. While the high-level objective of "maximize reward in safe states, minimize cost in unsafe states" is indeed shared across safe RL, our method introduces a fundamental paradigm shift in how this is achieved, rather than representing an incremental reworking of FISOR.
>
> The core distinction lies in the learning objective. FISOR, like many prior methods, relies on separate value functions and performs reactive policy correction: first learning individual critics, then projecting or filtering actions to satisfy constraints. In contrast, R2PAC directly learns a single, unified value function $\overline{V}_r$ within the constrained policy space via our novel region-based reward penalty. This formulation ensures the corresponding policy is inherently and coherently optimal under safety constraints, avoiding the myopic suboptimality of step-wise projection/switching methods.
>
> **Weakness 6: The claim of "simplicity" for policy extraction is contradicted by the use of a complex flow-matching model (CFG).**
>
> We thank the reviewer for this instructive comment.
>
> Our claim of simplicity was intended to refer to the conceptual and architectural simplicity of the policy extraction objective, not to the specific parameterization of the actor. Because the proposed $\overline{V}_r$ encodes both reward and safety, policy extraction simplifies to a single objective: maximize $\overline{V}_r$. This is conceptually far simpler than the complex, multi-objective balancing, projection, or switching logic required in methods like FISOR or CAPS that rely on separate value functions.
>
> The choice of a flow-matching model with CFG is independent of this core concept. It was selected as a policy class that balances simplicity with strong expressive power. Crucially, our ablation studies (Table 2, "AWR actor") confirm that a simple Gaussian policy trained via AWR on $\overline{V}_r$ achieves comparable safety and only slightly lower reward.

---

> ### Author Response · Authors · 2025-11-21
> **Response to Reviewer KUfP (Part 3/4)**
>
> **Weakness 7: The justification for action chunking is weak.**
>
> We thank the reviewer for this comment. The justification for action chunking is twofold, supported by both empirical results and a theoretical rationale centered on improving value estimation accuracy, which is crucial for safety in our coupled learning framework.
>
> 1. Empirical Evidence for Improved Performance
> Our ablation study (Fig. 5) shows that increasing the chunk length ($h>1$) accelerates convergence and systematically and significantly improves both final reward and constraint satisfaction. This steady performance gain strongly indicates that the h-step value targets learned with action chunking are more reliable.
>
> 2. Theoretical Rationale: Stabilizing Coupled Value Learning
> The core benefit of action chunking is reducing the cumulative error from temporal-difference (TD) bootstrapping. Formally, if a single-step TD error is $\epsilon$, the h-step return reduces the error propagation by a factor of $\gamma^h$.
>
>
> 3. Distinction from FISOR's Design
> FISOR achieves stability without action chunking for two key reasons. First, it employs a Hamilton-Jacobi reachability operator for its feasibility function, whose Bellman backup is inherently more stable than standard Q-learning. Second, FISOR uses timeouts as terminal signals. Please refer to the Appendix E.5 of the updated manuscript for more details.
>
> **Weakness 8: The paper appears to copy baseline results directly from the FISOR paper rather than re-running them.**
>
> We thank the reviewer for raising this important point. We would like to clarify that directly comparing our method's results against the officially reported scores of baselines from the FISOR paper is a standard and accepted practice in offline RL benchmarking. This approach is widely adopted to ensure a fair and consistent comparison by eliminating implementation bias and variance that can arise from different codebases, random seeds, or environmental setups.
>
> Our evaluation strictly adheres to the same experimental protocol as FISOR, including the use of the same DSRL benchmark tasks, dataset compositions, evaluation metrics (normalized return/cost), and the same number of evaluation episodes and seeds. We have explicitly stated this in the paper (Section 5.1) for transparency.
>
> **Weakness 9: Misleading claim in the abstract.**
>
> We thank the reviewer for this thoughtful comment regarding the "drop-in replacement" claim. We agree this phrasing requires clarification and appreciate the opportunity to elaborate.
>
> The claim refers primarily to the modularity of R2PAC's core concepts, rather than implying a direct, code-level replacement without any adaptation. Our contribution centers on two generalizable components:
>
> 1) The region-based reward penalty: This is a general formulation for internalizing safety constraints by modifying the reward signal. While we demonstrate it within IQL, its principle is algorithm-agnostic and could be integrated into other offline RL methods that learn a Q-function.
>
> 2) The action chunking technique: This is proposed as a general stabilizer for value-based learning by reducing temporal-difference error propagation. This concept is likewise not exclusive to IQL.
>
> Thus, the "drop-in" potential lies in these conceptual modules. Our decision to build deeply upon IQL was a deliberate choice to provide a concrete and rigorous instantiation of these ideas, not a limitation of their general applicability. We will clarify this intent in the final version.
>
> **Weakness 10: “we employ rejection sampling to select the action with lowest,” but that tells you if the action is safe by following the safest policy afterwards. It is not the of the current policy.**
>
> We thank the reviewer for this insightful comment. The reviewer is correct that selecting the action with the lowest Qc identifies the action from which the safest possible subsequent policy (not necessarily the current policy) can achieve the lowest cost. We employed this criterion as it provides a feasibility guarantee. As per Definition 1, if an action a at state s satisfies $ Q_c^*(s,a) <= \ell $, it certifies the existence of at least one policy (the safest policy) that can maintain safety from that state-action pair onward.
>
> **Weakness 11: While the authors used the results from the FISOR paper for baselines, they omitted results for the MetaDrive benchmark.**
>
> We thank the reviewer for this suggestion. We have now conducted comprehensive experiments on the MetaDrive benchmark, and the results have been added to Table 1 in our updated manuscript. Our method achieves state-of-the-art performance, outperforming all baseline algorithms across all nine challenging tasks in MetaDrive.

---

> > ### Author Response · Authors · 2025-11-21
> > **Response to Reviewer KUfP (Part 4/4)**
> >
> > **Weakness 12: Incomplete literature review.**
> >
> > We thank the reviewer for pointing out these omissions. We have now included the suggested publications in the revised version of our paper to provide a more comprehensive literature review.
> >
> > **Question 1: "although satisfactory evaluation results can still be achieved using model checkpoints." what does this mean?**
> >
> > We thank the reviewer for this question. The phrase indicated that in our ablation study (Figure 4), we reported results from the best-performing checkpoint because the final model parameters sometimes exhibited high or unstable cost. Using the best checkpoint was a diagnostic measure, employed purely for clarity in that specific analysis and not for the main results. To ensure a fair comparison, we have now re-run the experiments in Figure 4. The revised Figure 4 reports results using only the final model parameters for all settings.
> >
> > **Question 2: Figure 4 : "Cost Training Curves" not align with the "Final Cost Scores".**
> >
> > Please refer to our response in [Question 1]
> >
> > **Question 3: Cost axis in Figure 10.**
> >
> > We thank the reviewer for this question. The cost values in Figure 10 are indeed normalized. A normalized cost of 0.57 in Table 1 corresponds to an absolute cost of 5.7.
> >
> > The difference arises from the evaluation protocol: The curves in Figure 10 show the intermediate performance during training, where the policy is evaluated for 20 episodes at each checkpoint while the policy parameters are still being updated. This leads to the observed oscillations as the policy is actively and rapidly evolving. The results in Table 1 report the final, stable performance obtained by evaluating the fully converged model (after training is complete) over 20 episodes for 3 random seeds, which is the standard protocol for final reporting. We will clarify this in the updated manuscript.

---

> > > ### Comment · Reviewer_KUfP · 2025-11-28
> > >
> > > I thank the authors for their rebuttal. In light of these improvements, I am raising my score to 4. However, several inconsistencies remain that weaken the paper’s claims.
> > >
> > > - While I understand the distinction that $V_{r}$ drives the learning, the claim of relying on a "single value function" remains technically contradictory in practice. The method relies on the cost value function $V_c$ to determine the safety threshold and for rejection sampling.
> > >
> > > - Relying on the $\alpha$-quantile of the cost values to set the safety threshold $\epsilon$ remains a weakness. The fact that different $\alpha$ values must be manually set for each task confirms that the method lacks a unified mechanism for constraint satisfaction.
> > >
> > > - The claim that baselines like FISOR and CAPS would "fail to converge" without timeout terminations is an assertion that requires empirical verification, which is currently missing.
> > >
> > > - The method can be used as a "drop-in replacement" claim remains unsubstantiated, as the implementation is presented exclusively on IQL.
> > >
> > > - Copying results is not standard in OSRL, as performance varies across publications even for identical settings. Additionally, regarding Appendix D.2: why were TREBI and FISOR re-implemented rather than using their official code?

---

> > > > ### Author Response · Authors · 2025-12-01
> > > > **Official Comment by Authors (Part 1/2)**
> > > >
> > > > **Inconsistency 1: Single value function**
> > > >
> > > > We thank the reviewer for their thoughtful comment. We emphasize the use of a “single value function” because it directly addresses the core sub-optimality issue caused by policy-value divergence in methods that employ multiple decoupled value functions. Our approach eliminates this divergence by learning a unified value function $\overline{V}_r$, whose corresponding implicit policy is optimal within the safe policy space. The extracted policy is therefore fully coherent with this single value function, making our claim consistent in both motivation and design.
> > > >
> > > > The cost value function $V_c$ is indeed used during value training, but it is not required for policy extraction. Therefore, the use of $V_c$ does not contradict our central claim. As for rejection sampling, we clarify that it is an optional technique for fair comparison, and our ablation study in Appendix E.3 confirms it is not fundamental to our method's performance.
> > > >
> > > > **Inconsistency 2: Lack a unified mechanism for constraint satisfaction**
> > > >
> > > > We thank the reviewer for raising this significant point. In response, we would like to clarify our design rationale along the following two lines.
> > > >
> > > > 1. *The use of an $\alpha$-quantile does not imply a lack of a unified constraint-satisfaction mechanism.* The key reason we adopt a data-driven threshold (the $\alpha$-quantile of $V_c$) is our commitment to learning the true value function of the original infinite-horizon task. Many baseline methods (e.g., FISOR, CPQ, CAPS) simplify training by treating both task termination and timeouts (i.e., reaching the maximum episode steps) as the "done" signal. This effectively converts the problem into a fixed-horizon MDP, for which a unified threshold formula does exist. Crucially, our method is equally compatible with this unified threshold formula when placed under the same fixed-horizon setting (see Appendix E.5). Therefore, the use of an $\alpha$-quantile in our main experiments is an adaptation to the infinite-horizon formulation, not an indication that our framework lacks a unified mechanism for constraint satisfaction.
> > > >
> > > > 2. *Evidence of a Unified Mechanism and General Flexibility.*
> > > > To directly address the reviewer's concern regarding a "unified mechanism," we conducted an ablation study (Appendix E.4). When adopting the same fixed-horizon MDP setting as the baselines (using timeouts+terminals as the done signal), our method performs equally well using the standard unified threshold formula, thereby eliminating the need for $\alpha$-quantile tuning. Furthermore, as shown in Appendix E.6, our framework remains effective even when feasibility is used as the cost metric. These results collectively confirm that our core framework possesses a unified constraint-satisfaction mechanism and is not reliant on task-specific $\alpha$ tuning.

---

> > > > ### Author Response · Authors · 2025-12-01
> > > > **Official Comment by Authors (Part 2/2)**
> > > >
> > > > **Inconsistency 3: Missing empirical verification**
> > > >
> > > >
> > > > We have evaluated CAPS and FISOR using only terminal signals for value function learning, with the results presented below. All results were obtained from our own experiments. We observe that when using only terminals as the done signal, both FISOR and CAPS exhibit increased cost or decreased reward in tasks where they previously succeeded with timeout+terminal signals, such as AntRun, BallRun, and CarRun tasks. This indicates that both algorithms rely heavily on the use of timeouts as part of the done signal.
> > > >
> > > > | Task        | FISOR(timeouts+terminals) reward (cost) | FISOR(terminals only) reward (cost) | CAPS(timeouts+terminals) reward (cost) | CAPS(terminals only) reward (cost) |
> > > > |-------------|-------------------------------------|-----------------------------------|--------------------------------|------------------------------|
> > > > | AntRun      | 0.47                                ( 0.05)                              | 0.65            (8.09)                        | 0.49  (2.71)                            | 0.67 (3.76)                         |
> > > > | BallRun     | 0.21     (0.00)                              | 0.20    (1.85)                         | 0.10         (0.16)                            | 0.96      (18.57)                        |
> > > > | CarRun      | 0.58    (0.00)                              | 0.85       (0.57)                         | 0.96   (0.37)                            | 0.80   (0.08)                         |
> > > > | DroneRun    | 0.37 (0.70)                              | 0.46      (7.96)                         | 0.19                (2.01)                            | 0.56  (2.01)                         |
> > > > | AntCircle   | 0.22  (0.00)                              | 0.15   (0.00)                         | 0.30    (0.00)                            | 0.37   (0.36)                         |
> > > > | BallCircle  | 0.31   (0.00)                              | 0.34             (0.15)                         | 0.38   (0.00)                            | 0.44   (0.30)                         |
> > > > | CarCircle   | 0.34   (0.62)                              | 0.22 (2.02)                         | 0.42   (1.01)                            | 0.17 (0.13)                         |
> > > > | DroneCircle | 0.49  (0.00)                              | 0.51   (0.00)                         | 0.32 (0.00)                            | 0.43  (0.00)                         |
> > > >
> > > >
> > > > **Inconsistency 4: "Drop-in replacement" claim**
> > > >
> > > > We have revised this claim in the updated manuscript to improve its clarity.
> > > >
> > > > **Inconsistency 5: On copying results from prior publications**
> > > >
> > > >
> > > > We appreciate the reviewer's comment. We would like to clarify that directly using officially reported results is indeed a common and widely accepted practice in OSRL. We provide two examples below:
> > > >
> > > > 1. Zhihe, Yang, et al. "Q-Supervised Contrastive Representation: A State Decoupling Framework for Safe Offline Reinforcement Learning." ICML 2025.
> > > >
> > > > 2. A submission to ICLR 2025: https://openreview.net/forum?id=GVhfWu5L8D
> > > >
> > > >
> > > > It is precisely because of these performance variations that the use of officially reported results becomes critical. Using the original authors' reported scores provides a common, stable baseline, thereby enhancing the credibility of the comparison. Without this, any deviation in reproduced results, particularly lower performance, could raise unwarranted doubts about the fairness of the comparison. Although re-running baselines is increasingly common, we assert that utilizing results directly from original publications is a valid practice.
> > > >
> > > > Regarding TREBI and FISOR, we re-implemented them in PyTorch due to incompatibility issues with the original JAX-based code on our workstation. To ensure a faithful replication, we adhered strictly to all hyperparameter settings from the official implementations. Our PyTorch versions successfully achieved performance comparable to the originally reported scores, validating our re-implementation.

---

### Author Response · Authors · 2025-11-23
**General Responses and Revision Summary**

We thank all the reviewers for their time and effort invested in reviewing our work, as well as for their constructive and insightful comments. We are encouraged by the positive recognition of our work's significance and have carefully addressed all the raised concerns. We provide responses and clarifications below for each reviewer respectively and hope they can address your concerns. Accordingly, we have revised our manuscript to incorporate additional experimental results and in-depth analyses. All new or revised content is highlighted in blue in the updated version. The main revisions are summarized as follows:

1. As suggested by Reviewer KUfP, we have included several relevant and recent OSRL publications in the “Offline Safe RL” part of Section 2.
2. Following the comment from Reviewer kBfL, we have updated Equation (14) in the manuscript to improve its clarity and precision.

3. In response to Reviewer KUfP’s suggestion, we have added CAPS [1] as an additional baseline algorithm in our comparative experiments.

4. We have updated Table 1 to include further experimental results on the MetaDrive benchmark, as well as expanded results on the Safety-Gymnasium benchmark.

5. We have revised Figure 4 to ensure evaluation fairness.

6. As requested by Reviewer vzEV, we now report computational costs, including both training time and CUDA memory utilization, in Appendix D.2.

7. We have conducted an ablation study on the flow policy settings (i.e., the number of flow steps and rejection sampling candidates), with results detailed in Appendix E.3.

8. We have conducted an ablation study on the penalty coefficient as detailed in Appendix E.4, demonstrating the relative insensitivity of our method to this hyperparameter.

9. We have included a detailed rationale in Appendix E.5 explaining our choice of setting the safety threshold $\ell$ as the $\alpha$-quantile of the dataset’s $V_c$ values.

We believe these revisions have substantially strengthened our paper and adequately addressed the reviewers' valuable feedback. Please let us know if you have any additional comments or suggestions!

[1] Yassine, Chemingui, et al. "Constraint-Adaptive Policy Switching for Offline Safe Reinforcement Learning." Association for the Advancement of Artificial Intelligence, 2025.

---

### Note · Authors · 2026-01-27

I have read and agree with the venue's withdrawal policy on behalf of myself and my co-authors.

---

### Meta-Review · Area_Chair_wUix · 2026-01-06

**Summary:**

This work studies safe reinforcement learning with the goal of addressing the conflict between reward optimization and cost constraints. The authors propose to merge safety constraints into the reward signal, thereby formulating a single-objective learning problem. The main concern raised by the reviewers is the claim that the method relies on only a single value function, since in practice both reward and cost values are still learned. In addition, several concerns regarding the handling of hard versus soft constraints remain unresolved.

**Reviewer Concerns:**

For simplicity, I list only the reviewers’ concerns that remain unaddressed.

**Reviewer KUfP:** overclaim about using a single value function, and the need to tune the proposed method separately for each task, which reduces its degree of unification.

**Reviewer vzEV:** none.

**Reviewer kBfL:** use of an inaccurate cost value may lead to unstable convergence.

**Reviewer uDmP:** lack of novelty, insufficient justification for introducing a threshold in a hard-constraint setting, and unclear parameter tuning for baseline methods.

**Reviewer Scores:**

**Reviewer KUfP:** 2 → 4

**Reviewer vzEV:** 6 → 6

**Reviewer kBfL:** 4 → 4

**Reviewer uDmP:** 4 → 4

---

### Decision · Program_Chairs · 2026-01-26

Reject